# TAMPER-RESISTANT SAFEGUARDS FOR OPEN-WEIGHT LLMS

**Rishub Tamirisa**[∗†1,2,9]**, Bhrugu Bharathi**[∗1,3]**,**
**Long Phan**[9]**, Andy Zhou**[1,2]**, Alice Gatti**[9]**, Tarun Suresh**[1,2]**,**
**Maxwell Lin**[8]**, Justin Wang**[5,8]**, Rowan Wang**[6,8]**, Ron Arel**[1,2]**,**
**Andy Zou**[5,8,9]**, Dawn Song**[4]**, Bo Li**[2,7]**, Dan Hendrycks**[‡9]**, and Mantas Mazeika**[‡2,9]

[1]Lapis Labs,   [2]University of Illinois Urbana-Champaign,   [3]University of California, San Diego,
[4]University of California, Berkeley,   [5]Carnegie Mellon University,   [6]Harvard University,
[7]University of Chicago,   [8]Gray Swan AI,   [9]Center for AI Safety

## ABSTRACT

Rapid advances in the capabilities of large language models (LLMs) have raised widespread concerns regarding their potential for malicious use. Open-weight LLMs present unique challenges, as existing safeguards lack robustness to tampering attacks that modify model weights. For example, recent works have demonstrated that refusal and unlearning safeguards can be trivially removed with a few steps of fine-tuning. These vulnerabilities necessitate new approaches for enabling the safe release of open-weight LLMs. We develop a method, called TAR, for building tamper-resistant safeguards into open-weight LLMs such that adversaries cannot remove the safeguards even after hundreds of steps of fine-tuning. In extensive evaluations and red teaming analyses, we find that our method greatly improves tamper-resistance while preserving benign capabilities. Our results demonstrate that progress on tamper-resistance is possible, opening up a promising new avenue to improve the safety and security of open-weight LLMs.

## 1 INTRODUCTION

The most capable open-weight large language models (LLMs) released over the past year now rival closed-source frontier models (Llama Team, AI @ Meta, 2024). The availability of open-weight LLMs for anyone to download and use has yielded numerous benefits, including lowering costs for end users and enabling academic research on safety and security (Zou et al., 2023a). However, as these models become increasingly powerful, many have raised concerns that they could be repurposed by malicious actors to cause harm, motivating research on how to safeguard these models against malicious use.

Existing open-weight models often adapt safeguards designed for closed-weight models served through APIs (Touvron et al., 2023). These safeguards include refusal mechanisms and preference-based training, and they have provided substantial robustness against *input-based* jailbreaking attacks. However, recent work has demonstrated these safeguards are trivially defeated by attacks that edit model *weights*, breaking down after only a handful of fine-tuning steps (Qi et al., 2023). This poses a serious problem for open-weight models, because adversaries have full access to model weights and can tamper with built-in safeguards.

The vulnerability of open-weight models to tampering attacks poses risks for model developers as well. Under background tort law, AI developers must exercise reasonable care, meaning they have an obligation to take reasonable precautions to prevent foreseeable harm. If malicious actors can easily customize models to cause critical harm, model developers may inadvertently violate reasonable care

---

∗Equal contribution. ‡Equal advising. Correspondence to rishubt2@illinois.edu.
†Part of work done during an internship at the Center for AI Safety.

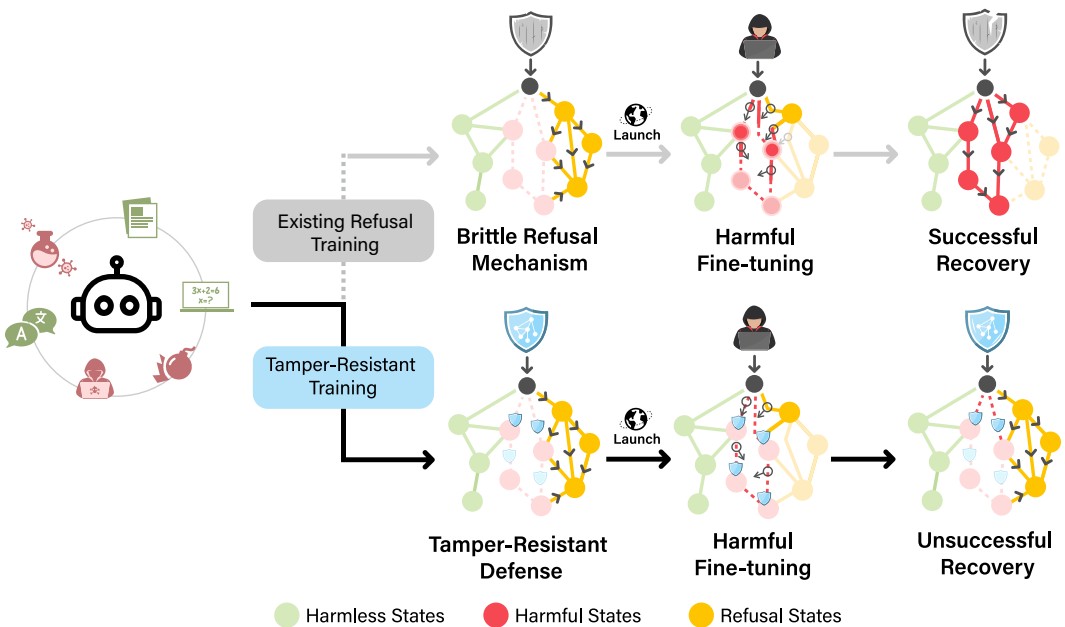

Figure 1: An illustration comparing two approaches to LLM safety when subjected to adversarial fine-tuning. The top branch shows conventional safeguards (like refusal training), which can be easily bypassed when adversaries fine-tune the model weights to remove safety constraints. The bottom branch demonstrates our proposed method TAR (Tampering Attack Resistance), which maintains robustness even when adversaries attempt to fine-tune the model to reintroduce harmful capabilities.

standards and become open to liability under existing law. Thus, there is an urgent need for more robust safeguarding techniques that can withstand tampering attacks.

In this work, we study the problem of tamper-resistant safeguards for LLMs. This problem is depicted in Figure 1. Unlike existing research on LLM safeguards, we focus on attacks that modify model weights, which we refer to as tampering attacks. This problem has been considered very challenging and by some intractable, as no method has yet provided substantial robustness to these attacks. However, making progress on this problem would provide a valuable tool to regulators and model developers by ameliorating the dual-use dilemma of open-weight models (Miller & Selgelid, 2007).

To demonstrate that progress on this problem is possible, we develop the first LLM safeguards that obtain strong robustness against a wide variety of tampering attacks. Our approach allows developers to add a safeguard such that tampering attacks cannot easily remove the safeguard, while preserving the general capabilities of the LLM. We achieve this by performing adversarial training against tampering attacks, leveraging approaches from meta-learning. We identify various crucial factors that enable our method to work, including the choice of tamper-resistance loss, the selection of train-time adversaries, and the two-stage approach that we use for building in safeguards.

We apply our method to develop tamper-resistant unlearning and refusal safeguards. In experiments, we demonstrate that our safeguards are far more robust to tampering attacks than prior methods. We stress-test our safeguards with extensive red teaming evaluations against 26 test-time adversaries, demonstrating resistance to fine-tuning attacks of hundreds of steps. We hope our results foster future work on this important problem. Our experiment code and models are available at https://github.com/rishub-tamirisa/tamper-resistance.

## 2 RELATED WORK

**Adversarial attacks on LLMs.** Due to the extensive pre-training distribution of modern LLMs, they are prone to generating harmful content (McGuffie & Newhouse, 2020; Sheng et al., 2019). To mitigate this, many LLMs undergo fine-tuning to implement safeguards (Bai et al., 2022; OpenAI, 2023; Touvron et al., 2023), using methods such as reinforcement learning from human feedback (RLHF) (Christiano et al., 2017; Ouyang et al., 2022) and direct preference optimization (DPO)

(Rafailov et al., 2023). While effective for normal use, these safeguards have been shown to be brittle, breaking down under jailbreak attacks (Jin et al., 2024a; Wei et al., 2023; Zou et al., 2023c) or a handful of fine-tuning steps on "uncensored" data (Qi et al., 2023; Yang et al., 2023; Zhan et al., 2023). This suggests current techniques for LLM alignment are inadequate, raising security concerns after deployment.

**LLM safeguards.** Since the discovery of these attacks, many safeguards have been proposed to defend against them. Against jailbreak attacks, defenses include system-level defenses Inan et al. (2023); Jain et al. (2023); Robey et al. (2023); Yuan et al. (2024); Zhou et al. (2024) that modify or filter model inputs or outputs and model-level defenses such as adversarial training Mazeika et al. (2024). Alternatively, some works explore machine unlearning as a way to remove harmful knowledge entirely with techniques such as influence functions (Bae et al., 2022; Koh & Liang, 2017), maximizing loss on forget sets (Eldan & Russinovich, 2023; Yao et al., 2023; Yu et al., 2023), or modifying representations (Belrose et al., 2024; Li et al., 2024; Sheshadri et al., 2024; Wu et al., 2023). However, jailbreaking defenses are not fully robust to adaptive adversaries Jin et al. (2024b); Liu et al. (2023), and existing unlearning methods are not robust to adversaries with access to model weights Lynch et al. (2024).

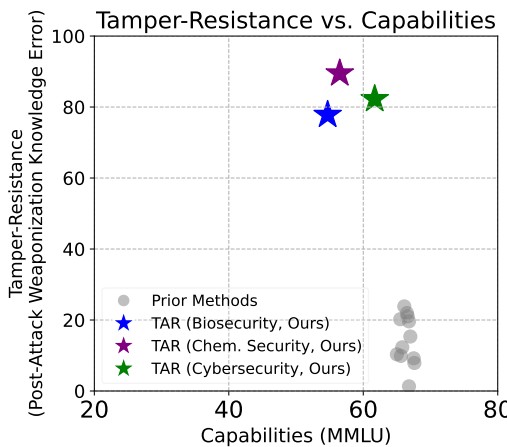

Figure 2: Comparison of our TAR method to 12 baseline safeguards. Unlike prior methods, TAR provides far greater tamper-resistance at similar levels of general capability, measured via MMLU. Tamper-resistance is computed as the normalized error on WMDP Biosecurity, Chemical Security, and Cybersecurity questions (Li et al., 2024), averaged across up to 26 fine-tuning attacks.

**Robust safeguards.** Several works have explored the tamper-resistance of unlearning methods for image classification (Golatkar et al., 2020a;b; Tarun et al., 2023). For bidirectional BERT-style models, Henderson et al. (2023) proposed a meta-learning approach for robustly preventing models from learning harmful tasks. In concurrent work, Deng et al. (2024) proposed a method extending this approach to small-scale vision classifiers and diffusion models. Recently, Liu et al. (2024) discussed the potential for robust unlearning in LLMs to improve the safety of open-source models, and Lynch et al. (2024) proposed evaluation metrics for robust unlearning in LLMs. To the best of our knowledge, no methods have been proposed for autoregressive LLMs that are robust to tampering attacks.

Several concurrent works have explored ways of defending LLM refusal mechanisms against fine-tuning (Huang et al., 2024a;c;b; Rosati et al., 2024a;b). Huang et al. (2024c) add a perturbation loss to make an LLM learn to produce embeddings that are more invariant to perturbations, Rosati et al. (2024a) maximize prediction loss on harmful generations while minimizing loss on refusals, and Rosati et al. (2024b) regularize harmful representations to look random. Unfortunately, these works evaluate against small sets of fine-tuning adversaries or have limited robustness. We corroborate this in our comparisons, finding that the approaches in the latter two works lack robustness to the tampering attacks in our evaluations.

## 3 TAMPER-RESISTANT SAFEGUARDS

### 3.1 THREAT MODEL

We assume the defender releases an LLM with weights $\theta_G$ and a safeguard $G$ applied. The defender's goal is to design $G$ such that $\theta_G$ obtains high values on `safety_metric`$(\theta_G)$ and `capabilities_metric`$(\theta_G)$. Moreover, the defender seeks to preserve a high value of `safety_metric`$(\theta_G)$ after the adversary's move. We consider a compute-bounded adversary with unrestricted access to $\theta_G$, enabling attacks that directly modify $\theta_G$. We refer to these as "tampering attacks." The adversary's goal is to obtain a model $\theta'_G$ that minimizes the safety metric given reasonable compute limits, such as fine-tuning for 500 steps. We note that in this work, we focus

solely on fine-tuning adversaries and not input-space "jailbreaking" adversaries. We assume the adversary will not spend a significant fraction of the compute required to pre-train the LLM, since at that point they could train their own model without safeguards.

## 3.2 Problem Definition and Metrics

We describe a general notation for quantifying the tamper-resistance of safeguards. Define $G$, $\theta_G$, `safety_metric`, and `capabilities_metric` as in the threat model. Let `attack` denote a compute-bounded adversarial attack that maps $\theta_G$ to $\theta'_G$, with stronger attacks obtaining lower values of `safety_metric`$(\theta'_G)$. We say that a safeguard $G$ is *tamper-resistant* if its post-attack `safety_metric`$(\theta'_G)$ is high across a broad range of strong test-time adversarial attacks $\mathcal{A}_{\text{test}}$.

Note that $\theta_G$ often modifies an underlying $\theta$ that lacks safeguards, often through a fine-tuning procedure. Additionally, strong tamper-resistance can be obtained if the safeguard simply overwrites $\theta$ with noise, but this model would no longer be useful. Thus, maintaining a high `capabilities_metric`$(\theta_G)$ is crucial, and evaluation of a safeguard must consider both its tamper-resistance and how well it preserves general capabilities.

We focus on two common safeguard domains: weaponization knowledge restriction and harmful request refusal. In each domain, we define safety and capabilities test metrics, which we use alongside test-time adversaries to evaluate tamper-resistant safeguards.

**Weaponization knowledge restriction.** In weaponization knowledge restriction, safeguards prevent the model from producing text about weaponization knowledge, while preserving capabilities for benign knowledge domains. Existing safeguards of this nature include representation engineering methods like circuit breaking (Zou et al., 2024). The `safety_metric` is defined as error on a *forget set*, and the `capabilities_metric` is defined as accuracy on a *retain set*. Specifically, we consider the problem of restricting biosecurity, chemical security, and cybersecurity knowledge, and evaluate the resulting model on the *Weapons of Mass Destruction Proxy (WMDP)* benchmark (Li et al., 2024). *WMDP* contains 3,668 multiple-choice questions, spanning biosecurity, chemical security, and cybersecurity knowledge. Importantly, *WMDP* questions do not evaluate hazardous knowledge directly, but instead measure proxy expert-level knowledge for each hazardous domain, such that restricting the expert-level knowledge would also restrict the hazardous knowledge. We define the forget set as the respective hazardous knowledge subject in *WMDP*, and retain set as the complement of the given subject in *MMLU* (Hendrycks et al., 2021), a multi-task question-answering benchmark spanning 57 tasks across a variety of knowledge domains.

**Harmful request refusal.** In the harmful request refusal setting, safeguards prevent the model from producing "harmful" outputs. We define the `safety_metric` as the complement of average Attack Success Rate (ASR) of various jailbreaking attacks, while the `capabilities_metric` captures the conversational abilities of $\theta_G$. Specifically, we use a static set of test cases from *HarmBench*, an automated red-teaming framework for measuring prompt jailbreak robustness in LLMs, to evaluate jailbreak ASR (Mazeika et al., 2024) after tampering attacks. We use *MT-Bench*, a multi-turn question-answering benchmark graded by an LLM judge, to evaluate conversational abilities (Zheng et al., 2023a).

## 3.3 Red Teaming

To properly measure the robustness of tamper-resistant safeguards, we conduct red-teaming with up to 26 adversaries, including many that are unseen at training time. In our evaluations, we subject our method to adversaries with varying compute budgets, access to held-out datasets, and diverse hyperparameters. For fine-tuning adversaries, we vary the learning rate, learning rate scheduler, optimization algorithm, and batch size. Many of these adversaries were fixed during early experiments, with some added over time as we found attacks that broke intermediate versions of our method. Extensive stress testing of this nature is critical for obtaining confidence in a tamper-resistant safeguard. For research on developing these safeguards, extensive red teaming also allows measuring incremental progress, using the number and strength of existing attacks one can defend against as a robustness metric.

---

**Algorithm 1** TAR: Tampering Attack Resistance

---

**Input:** Initial LLM parameters $\theta$, train-time adversary set $\mathcal{A}_{\text{train}}$, `capabilities_metric` proxy dataset $\mathcal{D}_{\text{retain}}$, `safety_metric` proxy dataset $\mathcal{D}_{\text{TR}}$, outer steps $N$, learning rate $\eta$, number of sampled adversaries $K$, tamper-resistance loss scale $\lambda_{\text{TR}}$, retain loss scale $\lambda_{\text{retain}}$, $h_\theta(\cdot)$ returns the residual stream hidden states for model parameters $\theta$

$\theta_0 \leftarrow$ **Apply Initial Safeguard to** $\theta$
**for** $i = 1$ **to** $N$ **do**
   $g_{\text{TR}} \leftarrow 0$ *# For accumulating tamper-resistance gradient*
   Sample $x_{\text{TR}} \sim \mathcal{D}_{\text{TR}}$
   **for** $k = 1$ **to** $K$ **do**
      Sample `attack` $\sim \mathcal{A}_{\text{train}}$
      *# Tamper-resistance loss from Equation 1*
      $g_{\text{TR}} \leftarrow g_{\text{TR}} + \frac{1}{K}\nabla_{\theta_{i-1}}\mathcal{L}_{\text{TR}}(\texttt{attack}(\theta_{i-1}), x_{\text{TR}})$
   **end for**
   Sample $x_r \sim \mathcal{D}_{\text{retain}}$
   *# RepE retain loss from Equation 2*
   $g_{\text{retain}} \leftarrow \nabla_{\theta_{i-1}}\Big(\mathcal{L}_{\text{LM}}(\theta_{i-1}, x_r) + \|h_{\theta_{i-1}}(x_r) - h_\theta(x_r)\|_2^2\Big)$
   *# Full tamper-resistance update*
   $\theta_i \leftarrow \theta_{i-1} - \eta\Big(\lambda_{\text{TR}} \cdot g_{\text{TR}} + \lambda_{\text{retain}} \cdot g_{\text{retain}}\Big)$
**end for**
$\theta_G \leftarrow \theta_N$
**return** $\theta_G$

---

# 4 SAFEGUARD TAMPER-RESISTANCE TRAINING

To obtain tamper-resistant safeguards, we propose a new method outlined in Algorithm 1 inspired by adversarial training and meta-learning to directly strengthen LLM safeguards against tampering attacks, called Tampering Attack Resistance (TAR). We identify unique properties of this adversarial training regime and leverage them to improve robustness.

Our method for training tamper-resistant safeguards consists of two phases: (1) model safeguarding and (2) tamper-resistance training.

## 4.1 MODEL SAFEGUARDING

The method begins by including an initial safeguard $G$ into a base model $\theta$. For example, initial safeguards for knowledge restriction can be drawn from a wide variety of existing methods, including circuit breaking (Li et al., 2024; Zou et al., 2024) or constrained gradient ascent for a particular knowledge domain. Similarly, we can include a refusal safeguard by performing RLHF (Ouyang et al., 2022) or DPO (Rafailov et al., 2023) on refusal completions. Importantly, these initial safeguards do not need to be tamper-resistant. Empirically, we find that this safeguarding step is crucial for preserving a low pre-attack `safety_metric`.

## 4.2 TAMPER-RESISTANCE TRAINING

Starting from $\theta_{G_0}$, we train the tamper-resistant $\theta_G$ using a novel adversarial training procedure. Namely, we train against a set of tampering attacks $\mathcal{A}_{\text{train}}$, where the defender's objective is to maximize a proxy `safety_metric` after applying an adversarial attack `attack` $\sim \mathcal{A}_{\text{train}}$ to $\theta$. Since it may not be feasible to differentiate through `attack`, we draw on insights from prior work in meta-learning, defining $\texttt{attack}(\theta_G) = \theta'_G = \theta_G + \texttt{attack}'(\theta_G)$ as a perturbation on top of initial parameters, where backpropagation through `attack`$'$ is approximated with a straight-through estimator (Bengio et al., 2013).

We focus on supervised fine-tuning (SFT) adversaries where `attack` applies several steps of optimization to $\theta_G$, which allows straight-through estimation through `attack`$'$ to benefit from the setting and approximations of first-order MAML (Finn et al., 2017). However, we note key differences in our approach from standard meta-learning and prior methods (Finn et al., 2017; Henderson et al.,

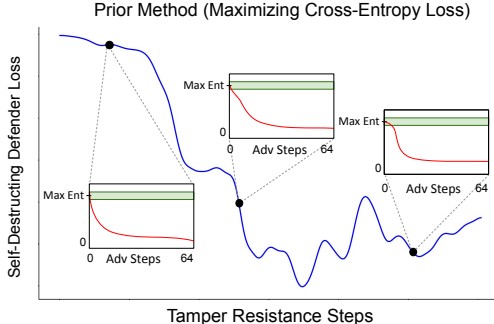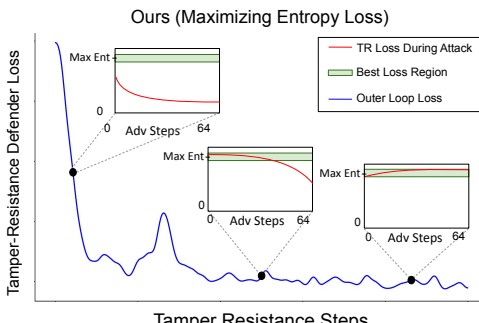

Figure 3: The choice of tamper-resistance loss is crucial for obtaining good performance. Here, we show loss trajectories when the tamper-resistance loss is negative cross-entropy (left), versus negative entropy (right), over the course of TAR for 750 steps. Outer loop losses (blue) are reduced by the defender, and inner-loop losses (red) are reduced by the train-time adversary. When the tamper-resistance loss maximizes cross-entropy (left), the adversary is only affected earlier in its trajectory and quickly recovers. By contrast, when the tamper-resistance loss maximizes entropy (right), the inner loop adversary is eventually thwarted along its entire trajectory. Plots are smoothed.

2023). In particular, traditional meta-learning techniques seek to obtain a model initialization that is *close* to optimality on multiple test distributions. In our setting, we seek to obtain an initialization that is *far* from optimality on multiple adversaries' test distributions. Novel to our approach in this new setting is the use of a tamper-resistance loss in the "outer loop" that differs from the fine-tuning adversary's loss function and serves to maximize the proxy safety metric. We depict this structure in Algorithm 1, and explain the objective below.

**Impeding the adversary's loss.**   The aim of tamper-resistance training is to prevent adversaries with large compute budgets from reducing the `safety_metric` at *test-time*. In adversarial training for tamper-resistance, we define a tamper-resistance loss $\mathcal{L}_{\text{TR}}$ that counters `attack`. We operationalize our goal of *avoiding adversary optimality* as searching for $\theta$ such that $\mathcal{L}_{\text{TR}}$ is minimized for `attack(`$\theta$`)`.

Empirically, we find that the choice of tamper-resistance loss $\mathcal{L}_{\text{TR}}$ significantly affects this goal. Prior work (Henderson et al., 2023) negates the loss of a fine-tuning adversary, in which the aim is to arbitrarily maximize the adversary's loss throughout fine-tuning. This formulation has two issues: (1) maximizing a cross-entropy loss can cause divergence; (2) empirically we observe that when using this objective against fine-tuning adversaries, the model learns to explode the adversary's loss for the first few inner loop steps, while loss at later steps remains low. In Figure 3, we show the difference in choosing $\mathcal{L}_{\text{TR}}$ to be a clamped negative cross-entropy loss vs. negative entropy loss for weaponization knowledge restriction. For the latter, $\mathcal{L}_{\text{TR}}$ is eventually satisfied for all inner loop steps. For harmful request refusal, we choose $\mathcal{L}_{\text{TR}}$ to be the DPO loss (Rafailov et al., 2023). We provide further detail on the choice of $\mathcal{L}_{\text{TR}}$ in both settings in Appendix C.3, as well as an extended depiction of the test-time loss characteristics in Appendix D.3 and Figure 6.

**Tamper-resistance objective.**   We now describe the general proxy objective used for preventing a tampering attack from recovering weaponization knowledge or harmful behavior. For a given `safety_metric`, let $\mathcal{D}_{\text{TR}}$ and $\mathcal{L}_{\text{TR}}$ respectively be a dataset and loss function such that minimizing $\mathcal{L}_{\text{TR}}(\theta_G, \mathcal{D}_{\text{TR}})$ serves as a proxy objective for maximizing the `safety_metric(`$\theta_G$`)`. We define $\mathcal{D}_{\text{retain}}$ and $\mathcal{L}_{\text{retain}}$ correspondingly for `capabilities_metric(`$\theta_G$`)`. The defender's objective is to solve the following optimization problem:

$$\min_{\theta} \lambda_{\text{TR}} \cdot \mathbb{E}_{\texttt{attack} \sim \mathcal{A}_{\text{train}}} \big[ \mathcal{L}_{\text{TR}}(\texttt{attack}(\theta); \mathcal{D}_{\text{TR}}) \big] + \lambda_{\text{retain}} \cdot \mathcal{L}_{\text{retain}}(\theta; \mathcal{D}_{\text{retain}}), \qquad (1)$$

where $\mathcal{L}_{\text{TR}}$ is a tamper-resistance loss that counters `attack(`$\theta$`)`. The $\mathcal{L}_{\text{retain}}$ term is a representation engineering (Zou et al., 2023a) inspired retain loss for preserving performance on the capabilities proxy dataset $\mathcal{D}_{\text{retain}}$, given by

$$\mathcal{L}_{\text{retain}}(\theta; \mathcal{D}_{\text{retain}}) = \mathbb{E}_{x \sim \mathcal{D}_{\text{retain}}} \Big[ \mathcal{L}_{\text{LM}}(\theta, x) + \| h_{\theta}(x) - h_{\theta_{G_0}}(x) \|_2^2 \Big] \qquad (2)$$

where $h_{\theta}(\cdot)$ returns the residual stream hidden states for model parameters $\theta$ and $\mathcal{L}_{\text{LM}}$ is the standard language modelling cross-entropy loss. Empirically, we find that pushing retain-set residual stream

| Weaponization Domain | Model | Pre-Attacks | | Post-Attacks (Avg) |
| --- | --- | --- | --- | --- |
| | | Retain ($\uparrow$) | Forget ($\downarrow$) | Forget ($\downarrow$) |
| Biosecurity | Random | 25.0 | 25.0 | 25.0 |
| | No Defense | 67.3 | 70.5 | 70.5 |
| | Max Entropy | 65.0 | 33.2 | 65.8 |
| | Min Posterior | 65.6 | 50.4 | 66.0 |
| | LLMU | 65.5 | 29.9 | 61.3 |
| | RMU | 65.8 | 31.2 | 64.9 |
| | TAR (Ours) | 54.7 | **28.1** | **35.2** |
| Chemical Security | Random | 25.0 | 25.0 | 25.0 |
| | No Defense | 68.2 | 47.8 | 47.8 |
| | Max Entropy | 67.5 | 50.0 | 45.7 |
| | Min Posterior | 66.8 | 49.5 | 47.5 |
| | LLMU | 67.0 | 30.1 | 44.3 |
| | RMU | 67.6 | **27.5** | 46.0 |
| | TAR (Ours) | 56.5 | 28.4 | **27.1** |
| Cybersecurity | Random | 25.0 | 25.0 | 25.0 |
| | No Defense | 68.2 | 46.4 | 46.4 |
| | Max Entropy | 66.5 | 28.7 | 41.7 |
| | Min Posterior | 66.6 | 41.8 | 41.9 |
| | LLMU | 66.1 | 27.6 | 41.3 |
| | RMU | 66.8 | 29.5 | 42.2 |
| | TAR (Ours) | 60.7 | **23.6** | **28.6** |

Table 1: Pre-Attack and average Post-Attack accuracies for WMDP Biosecurity, Chemical Security, and Cybersecurity for TAR and all other baselines, reported for Llama-3-8B. The average Post-Attack accuracy is computed as the average accuracy across the 26 fine-tuning attacks discussed in Section 5, averaged over multiple seed repeats. TAR is the only method that maintains low Post-Attack recovery while preserving high Retain MMLU and low Forget accuracies. All values are percentages.

representations to be close to the base model $\theta_{G_0}$ via the $\ell_2$-norm loss in Equation 2 maintains a high `capabilities_metric`$(\theta_G)$. In Equation 1 we include $\lambda_{\text{TR}}$ and $\lambda_{\text{retain}}$ as scalar weightings for the tamper-resistance loss and retain loss, respectively. We provide further details on the design of the tamper-resistance loss function in Appendix C.3 as well as an efficiency trick for sampling fine-tuning attacks for TAR in Appendix C.4.

## 5 EXPERIMENTS

We evaluate TAR in weaponization knowledge restriction and harmful request refusal settings, with results shown in Table 1 and Table 2 respectively. We discuss the setup, baselines, and analysis for our results. In each setting, we use a specific set of training adversaries $\mathcal{A}_{\text{train}}$ and test adversaries $\mathcal{A}_{\text{test}}$. Further experiment details are presented in Appendix E.

### 5.1 WEAPONIZATION KNOWLEDGE RESTRICTION

We now describe the setup, baselines, and results for our weaponization knowledge restriction experiments, including the knowledge domains, optimizers, and evaluation details.

**Setup.** We focus on implementing tamper-resistant safeguards for restricting proxy weaponization knowledge about biosecurity, chemical security, and cybersecurity from Llama-3-8B-Instruct (AI@Meta, 2024) that has been initially safeguarded via the *Random Mapping* method discussed in Appendix C.2. For each weaponization domain, we assign $\mathcal{D}_{\text{TR}}$ to the corresponding forget set described in Appendix E.1. We proceed to sample train-time 64-step fine-tuning attacks from different data distributions, detailed in Appendix E.2. We use $N = 750$ outer loop steps, ScheduleFree AdamW (Defazio et al., 2024) with a learning rate of $2 \times 10^{-5}$ as the outer loop tamper-resistance optimizer. For biosecurity and cybersecurity we set the tamper-resistance loss scale $\lambda_{\text{TR}}$ to 4.0, and use $\lambda_{\text{TR}} = 3.0$ for chemical security. We use $\lambda_{\text{retain}} = 1.0$ in all settings. Lastly, we evaluate

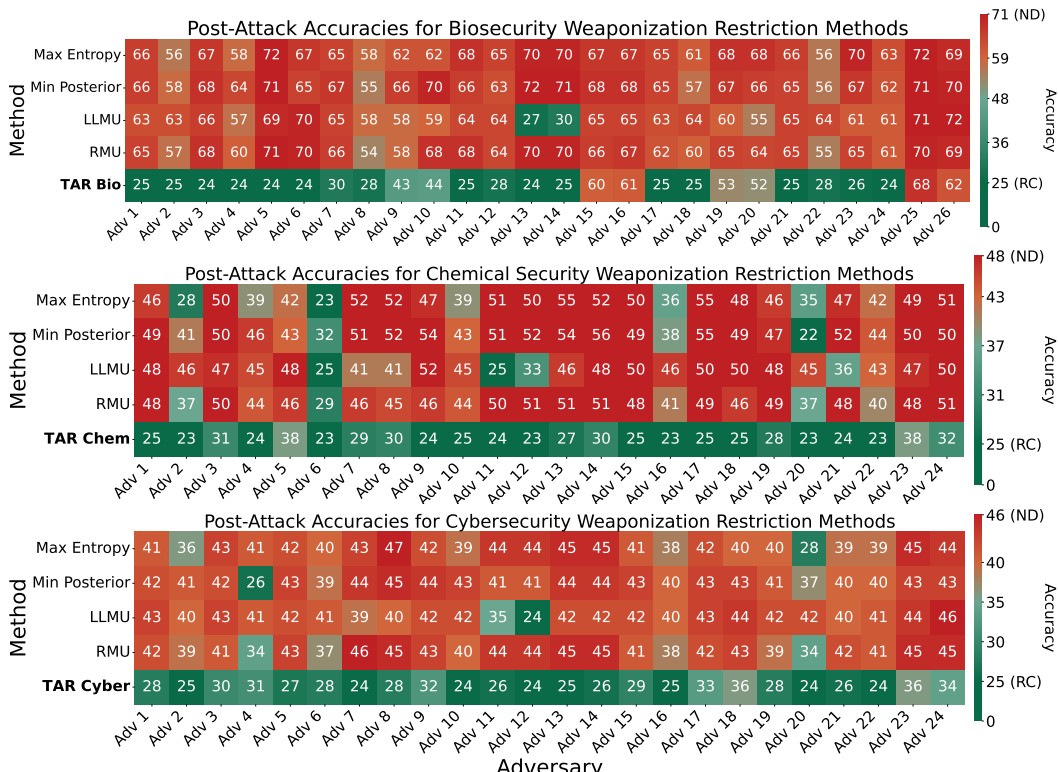

Figure 4: Red teaming results across weaponization domains. Values show percentages, with Random Chance (RC) at 25% and "ND" indicating No Defense WMDP scores. Red indicates attack performance approaching No Defense levels. We evaluate each defense against a diverse range of strong adversaries described in Appendix F.1). Accuracies are reported as averages over 3 repeats of each attack with different seeds. Compared to prior safeguards, TAR greatly increases tamper-resistance for nearly all adversaries.

Pre-Attack and Post-Attack accuracy on corresponding WMDP subjects (Li et al., 2024) averaged across all adversaries in Appendix F.1, and measure benign capabilities via the complement of subjects related to each proxy weaponization domain in MMLU (Hendrycks et al., 2021).

**Baselines.** We evaluate two recently proposed knowledge restriction methods: *RMU* (Li et al., 2024) and *LLMU* (Yao et al., 2023). We also design two baseline methods for knowledge restriction: *Min Posterior*, which minimizes posterior loss on forget set tokens; *Max Entropy*, which maximizes entropy on forget set tokens. Two additional methods, *MLAC* (Henderson et al., 2023) and *SOPHON* (Deng et al., 2024), require substantial modifications for the LLM setting, so we show results on adapted versions of these baselines in Appendix G.3.

**Results.** We show weaponization knowledge restriction safeguard results on Llama-3-8B-Instruct in Table 1 and Figure 2. These results are averaged across all adversaries described in Appendix F.1. Our large-scale experiments corroborate the findings in recent work that existing LLM safeguards are extremely brittle to fine-tuning attacks. By contrast, TAR maintains low post-attack forget accuracy across all three domains. However, we observe that TAR lowers retain accuracy by 10.6% on average, indicating a trade-off between benign capabilities and robustness. In Figure 4, we observe that TAR is robust to significantly more fine-tuning attacks than all prior methods. While existing baselines break down under most attacks, TAR obtains a post-attack forget accuracy near random chance for nearly all attacks, indicating a successful defense.

Overall, we find that TAR provides significantly more robustness to realistic fine-tuning attacks than all prior methods, including SFT attacks that utilize completely held-out data. We also include further analysis of TAR's test-time loss behavior in Appendix D.3, in which we empirically observe TAR's convergence and robustness. These results demonstrate for the first time that obtaining strong tamper-resistance for open-weight LLMs may be possible.

|                          | Refusal Trained | R2D2 | RepNoise | RR   | TAR (Ours) |
|--------------------------|-----------------|------|----------|------|------------|
| Pre-Attacks MT-Bench (↑) | 8.1             | 6.0  | 6.2      | 8.0  | 6.3        |
| Avg. Post-Attacks ASR (↓)| 72.5            | 78.3 | 74.5     | 84.8 | **63.9**   |

Table 2: Average Post-Attack HarmBench ASR, reported for TAR, Representation Rerouting (RR), and the Refusal Trained Llama-3-8B-Instruct model across 5 fine-tuning attacks depicted in Appendix F.2, as well as Pre-Attack MT-Bench. TAR is more robust than other methods after tampering, while maintaining comparable MT-Bench performance. ASR values are percentages.

## 5.2 HARMFUL REQUEST REFUSAL

We now describe the setup, baselines, and results for our harmful request refusal experiments, including the datasets used and evaluation details.

**Setup.** For harmful request refusal training, we seek to make existing refusal safeguards in Llama-3-8B-Instruct robust to tampering attacks. We sample train-time adversaries that perform 64-step SFT attacks using the Anthropic-HH-RLHF dataset (Bai et al., 2022), following the methodology in Appendix E.2. Similar to the weaponization knowledge restriction setting, we use $N = 100$ outer loop steps, ScheduleFree AdamW (Defazio et al., 2024) with an LR of $6 \times 10^{-5}$ as the outer loop tamper-resistance optimizer, and loss scales of $\lambda_{\text{TR}} = 0.1$, $\lambda_{\text{retain}} = 1.0$. We evaluate the Post-Attack jailbreak attack success rate (ASR) on HarmBench (Mazeika et al., 2024) after the tampering attacks in Appendix F.2, and measure benign capabilities preservation via MT-Bench (Zheng et al., 2023b), which evaluates multi-turn conversation ability.

**Baselines.** We consider 4 baselines alongside our TAR model: Llama-3-8B-Instruct *(Refusal Trained)*; *Representation Rerouting (RR)* (Zou et al., 2024) on Llama-3-8B-Instruct, which trains to push representations for harmful inputs to be orthogonal to the original representations in Llama-3-8B-Instruct; *R2D2* (Mazeika et al., 2024) on Zephyr-7B (Tunstall et al., 2023), which performs adversarial training against GCG attacks (Zou et al., 2023b); and *RepNoise* (Rosati et al., 2024b) on Llama-2-7B (Touvron et al., 2023), which regularizes harmful representations to noise.

**Results.** We show refusal results in Table 2. While the Refusal Training, RR, and R2D2 baselines resist jailbreak attacks in HarmBench before tampering, we find that percentage attack success rate jumps up to above 77 after tampering, while our TAR method only rises to 61.7. Since we apply our TAR refusal safeguard to Llama-3-8B, it does reduce MT-Bench by 1.7. However, this exceeds the MT-Bench score of fairly capable open-weight models, indicating that benign capabilities are largely preserved. We leave the exploration of the full impact on capabilities to future work. Additional results are in Table 11. In general, we find that our TAR model refuses more Post-Attack jailbreaks than previous baselines, and demonstrates the flexibility of the tamper-resistance objective to accommodate the harmful request refusal setting.

## 5.3 ANALYSIS

**Red teaming.** To assess the tamper-resistance of our models, we conduct an extensive suite of supervised fine-tuning attacks with 26 distinct adversaries in the Biosecurity setting and 24 distinct adversaries in the Chemical Security and Cybersecurity settings. We vary the optimizer, number of optimization steps, learning rate, learning rate schedule, fine-tuning dataset, batch size, and overall fine-tuning method (e.g., full fine-tuning versus parameter-efficient fine-tuning). By default, our attacks use 500 fine-tuning steps. Full details for these adversaries are provided in Table 9.

We show red teaming results in Figure 4. While baseline safeguards withstand fine-tuning attacks in a small number of cases, most adversaries succeed in removing the safeguards. By contrast, our TAR safeguard is robust to a wide range of adversaries. This shows that tamper-resistance is a tractable problem on which progress can be made. However, we find our method exhibits varying robustness to parameter-efficient fine-tuning (PEFT) and some out-of-distribution LR attacks, highlighting the current sensitivity of the method to the adversary distributions sampled during TAR training. These findings reinforce the importance of extensive red teaming when developing tamper-resistant defenses to reveal the scope of their protection. We hypothesize that future work could easily address these limitations, as we demonstrate in Appendix D.2 that targeted patching of vulnerabilities is possible.

As mentioned in Section 3.1, the threat model we consider involves SFT weight tampering adversaries and not input-space "jailbreaking" adversaries, nor does TAR explicitly optimize for input-space robustness (Mazeika et al., 2024; Zou et al., 2023c; 2024). Nonetheless, we find that TAR's pre-attack forget accuracies are comparable to baselines in Table 1, and believe that explicitly defending against input-space attacks alongside tampering attacks would be a good direction for future work.

## 6 CONCLUSION

We introduced a novel method for implementing tamper-resistant safeguards for LLMs and explored applications in weaponization knowledge restriction and harmful refusal training. We compare our results to prior work in each setting, finding that our method is the first method robust under the rigorous red-teaming evaluation that we consider. More broadly, we demonstrate that progress on open-weight tamper-resistance is tractable. We believe this line of research is crucial for enabling ongoing deployment of robust, open-weight LLMs, ensuring their alignment with regulatory frameworks and preemptively addressing the risk of malicious use.

**Acknowledgements.**   We thank Steven Basart and Luis Fernandez for providing valuable feedback for the paper, as well as Xiangyu Qi, Boyi Wei, Nicholas Carlini, Prateek Mittal, and Peter Henderson for useful discussions. We also thank Andriy Novykov and the Center for AI Safety for providing significant compute resources for this project, as well as Volodymyr Kindratenko and the National Center for Supercomputing Applications (NCSA) and Illinois Campus Cluster Program (ICCP) for supporting our computing needs. This work used NVIDIA GPUs at NCSA Delta through allocations CIS230117 from the Advanced Cyberinfrastructure Coordination Ecosystem: Services & Support (ACCESS) program, which is supported by NSF Grants #2138259, #2138286, #2138307, #2137603, and #2138296.

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

## A    LIMITATIONS

Our method for training tamper-resistant safeguards demonstrates considerable robustness against a wide range of tampering attacks, yet several avenues for improvement remain: (1) While we focus on supervised fine-tuning attacks, the broader spectrum of open-weight tampering techniques necessitates diverse future red-teaming efforts. (2) Scaling to larger models poses computational challenges that require optimization to reduce overheads.

Additionally, in cases where TAR maintains a low post-attack forget accuracy, the post-attack retain accuracy is also low. By contrast, we found in preliminary experiments that post-attack retain accuracy for many of the baselines remained high. We note that this is acceptable because post-attack retain performance is not of concern to the defender; rather, the responsibility falls on the attacker to preserve it after tampering.

However, this does mean benign users trying to fine-tune the model must ensure their data is not contaminated by forget set data, lest their fine-tuned model have poor performance. This could make the method harder to use in practice. Thus, maintaining high post-attack retain accuracy would be a useful direction for future work to explore.

Tamper-resistance alone cannot fully mitigate the risks of malicious AI use. While it raises the initial costs for adversaries, it can eventually be circumvented. Once open-weight models are released, they cannot be "unreleased," leaving any compromised defenses permanently vulnerable. Therefore, tamper-resistance should be considered a supplement to the broader effort of of improving the offense-defense balance of AI systems. Addressing these limitations will improve the robustness of LLMs to tampering and better support open-weight model developers.

## B    AUTHOR CONTRIBUTIONS

Rishub Tamirisa, Bhrugu Bharathi, and Mantas Mazeika led the project, including developing methods and contributing writing for all sections of the paper. Rishub Tamirisa implemented most of the code for the method and baselines, and Bhrugu Bharathi implemented most red-teaming evaluations in the paper. Rishub and Bhrugu carried out all main experiments, and orchestrated analysis experiments for Maxwell Lin and Tarun Suresh to conduct. Rowan Wang and Justin Wang curated datasets for the main experiments for the final version of the method. Long Phan and Alice Gatti implemented red-teaming evaluations, contributed figures, and gave feedback for writing drafts. Andy Zhou ran experiments for robust refusal, contributed writing the Related Work section, and helped create figures. Ron Arel organized compute logistics from the NCSA, and provided personnel for experiments. Andy Zou provided personnel for experiments and high-level advising. Dawn Song and Bo Li provided high-level advising for the project. Dan Hendrycks provided substantial advising for framing and writing, as well as method suggestions. Dan Hendrycks also provided significant compute resources from the Center for AI Safety. Mantas Mazeika was the primary project advisor during the majority of its duration, and was involved in most decisions regarding the method, experiments, and writing.

## C  METHOD DETAILS

### C.1  NOTE ON UPDATED IMPLEMENTATION

After the initial release of this paper, we identified a data contamination issue in which our instruction-tuning retain dataset, Magpie-Align (Xu et al., 2024), contained a significant amount of forget set content. This resulted in an unintended input-space vulnerability (Qi et al., 2024). After cleaning the dataset and re-training new TAR models with the same methodology and hyperparameters, the input-space vulnerability is no longer observed (Qi et al., 2024).

Additionally, Qi et al. (2024) found that TAR's robustness at 1,000 of steps of fine-tuning varies when changing the dataloader shuffling order, which we corroborate in Figure 5. After further investigation, we found an issue in the HuggingFace distributed dataloading sampler, where the HuggingFace sampler overrides user-defined seeds with a default seed. This resulted in the test-time dataloader shuffling orders being a subset of train-time shuffling orders in our initial release. Upon correcting the issue, we observe the variability in Figure 5.

However, while the loss plateau and robustness at 1,000 steps can vary with different dataloader shuffling orders, we do observe in Figure 5 and in nearly all adversaries that the loss plateau and robustness we discuss throughout the paper remains consistent across multiple shuffling orders for 500 steps of fine-tuning. We report these results in our updated Table 1, Figure 2,

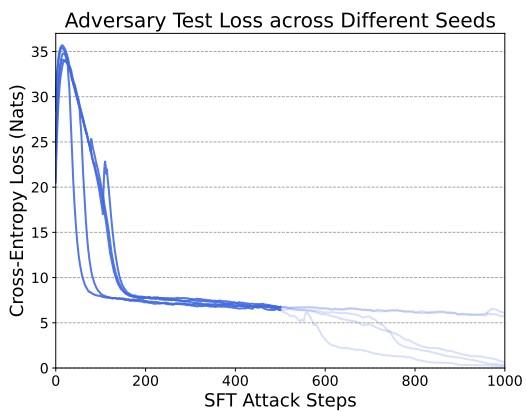

Figure 5: Test loss for five repeats of a 1,000-step SFT attack against a TAR-Bio safeguard, each using a different dataloader shuffling seed. TAR yields a consistent loss plateau for 500 steps, followed by a loss region of increased variability.

and Figure 4. All depicted per-adversary post-attack accuracies in Table 1, Figure 2, and Figure 4 are averaged over three replicates, each using a different random seed. We include further discussion on the loss plateau as well as concrete evidence of TAR's convergence in Appendix D.3. We do emphasize that TAR models we consider in our experiments optimize against SFT adversaries that use only $K = 64$ optimization steps yet achieve significant generalization to test-time adversaries using substantially more compute, greatly improving tamper-resistance compared to prior work.

### C.2  INITIAL WEAPONIZATION KNOWLEDGE RESTRICTION SAFEGUARD

Prior to tamper-resistance training, we install a safeguard that achieves surgical knowledge restriction on the target hazardous domain. Let $h_\theta(\mathcal{D})$ denote the distribution of post-decoder layer residual stream activations for input sequences sampled from some data distribution $\mathcal{D}$ and model weights $\theta$. We define $\texttt{rand\_hashed}(x)$ for some input sequence $x$, which returns fixed Gaussian-sampled vectors that are chosen via hashing the corresponding input token for each residual stream index of $x$ in $\theta$. As a proxy for scrubbing target representations according to downstream task labels, we propose a weaponization knowledge restriction safeguard termed *Random Mapping*, which maps $h_\theta(\mathcal{D}_{\text{TR}})$ to random noise as follows:

$$\min_\theta \mathbb{E}_{x \sim \mathcal{D}_{\text{TR}}}\left[1 - \left|\frac{h_\theta(x) \cdot \texttt{rand\_hashed}(x)}{\|h_\theta(x)\|\|\texttt{rand\_hashed}(x)\|}\right|\right] + \mathcal{L}_{\text{LM}}(\theta; \mathcal{D}_{\text{retain}}) \tag{3}$$

The objective of Equation 3 maximizes cosine similarity between row vectors in the residual stream in every layer of the LLM from $h(\mathcal{D}_{\text{TR}})$ and the hashed random vectors from $\texttt{rand\_hashed}(\cdot)$. By providing each token's residual stream a unique random vector to push toward, the loss encourages a "re-mapping" of token representations from $\mathcal{D}_{\text{TR}}$ to the noised vectors. We include an additional term for preserving performance on $\mathcal{D}_{\text{retain}}$ via the language-modelling cross-entropy loss $\mathcal{L}_{\text{LM}}$. We show the performance of the raw *Random Mapping* safeguard as an ablation in Table 5, listed as "Excl. Adv. Training."

### C.3 Designing the Tamper-resistance Loss

**Weaponization knowledge restriction.** For weaponization knowledge restriction, we summarize our intuition for ideal tamper-resistance loss design, corroborated by our empirical findings in Figure 3: *we seek to flatten the adversary's loss at a high value, rather than simply raise its y-intercept.*

We choose the tamper-resistance loss $\mathcal{L}_{TR}$ as an entropy loss to be maximized during the adversary's cross-entropy fine-tuning trajectory, since maximizing entropy would impede the adversary's cross entropy loss from decreasing during fine-tuning. In other words, we wish to obtain $\theta$ such that *after* an adversary performs a fine-tuning attack on $\theta$ via a cross-entropy loss, entropy is still high. We find that this formulation achieves the desired flattening behavior, and we depict the difference in flattening between the choosing $\mathcal{L}_{TR}$ to be a negative cross-entropy loss and negative entropy loss in Figure 3. In the lefthand plot, where $\mathcal{L}_{TR}$ is a cross-entropy loss, loss only increases in the first inner loop step. In the righthand plot, where $\mathcal{L}_{TR}$ is a negative entropy loss, entropy is eventually maximized in all inner loop steps. Figure 6 also demonstrates the generalization of the flat adversary loss behavior beyond the length of the simulated fine-tuning trajectories during TAR.

**Harmful request refusal.** For harmful request refusal, we choose $\mathcal{L}_{TR}$ to be the DPO loss (Rafailov et al., 2023), which works as follows. Given a DPO dataset containing pairs of rejected and refusal completions, the sampled `attack` performs SFT on rejected completions, and the tamper-resistance loss $\mathcal{L}_{TR}$ is a DPO loss computed on the pair chosen and rejected completions on parameter coordinates along the `attack` trajectory. This encourages TAR to find an initialization $\theta$ such that after a harmful fine-tuning attack, the model still prefers refusal completions over harmful completions when given an harmful prompt. While this does not necessarily encourage a flat adversary loss, we find empirically in Table 2 that this formulation increases the average `safety_metric`$(\theta_G)$ defined in Section 3.2 after fine-tuning attacks on harmful data, detailed in Section 5.

We also observe that the length of the simulated adversary SFT trajectory during training affects test-time generalization in both Figure 6 and Appendix D.3. In particular, larger values of $K$ result in increased tamper-resistance for longer SFT attacks. However, to maintain reasonable runtime efficiency, we need a more efficient sampling technique than simply running $K$ independent trajectories of varying length for every outer-loop step in Algorithm 1, which we describe in Section C.4.

### C.4 Efficiently Sampling Fine-tuning Attacks

Optimizing Equation 1 with gradient descent requires simulating $K$ tampering attacks for each tamper-resistance optimizer update, which is prohibitively expensive to run when the sampled `attack` performs SFT and $\theta$ contains billions of parameters. Inspired by prior work on snapshot ensembles Huang et al. (2017), we leverage an efficiency trick: we can reuse the coordinates along steps of a single adversary fine-tuning trajectory of length $K$ to obtain $K-1$ additional (though non-independent) trajectories of increasing length. Using this trick, we collect all $K$ parameter coordinates along the trajectory into a single batch for computing the tamper-resistance losses, effectively sampling `attack` from $\mathcal{A}_{train}$ non-IID. To further improve runtime efficiency, we do not compute the tamper-resistance loss $\mathcal{L}_{TR}$ on all $K$ steps and instead sub-sample coordinates along the trajectory for computing $\mathcal{L}_{TR}$ within an adversary batch, for example every 4 adversary optimization steps. Additionally, we reduce variance in the tamper-resistance gradient by computing the tamper-resistance loss at each inner loop step on the same held-out batch, denoted as $x_{TR}$ in Algorithm 1.

### C.5 Implementation Details and Resource Requirements

We perform TAR training on Llama-3-8B-Instruct (Llama Team, AI @ Meta, 2024) with 8 NVIDIA 80GB A100 GPUs, leveraging distributed training via FSDP (Ren et al., 2021; Rajbhandari et al., 2020; Zhao et al., 2023). We use ZeRO Stage 3 from DeepSpeed (Rajbhandari et al., 2020), which shards optimizer states, gradients, and parameters during training. While the efficiency trick in Appendix C.4 improves runtime, we note additional considerations for conserving GPU memory.

First, simulating fine-tuning attacks that require additional state (e.g., momentum) in the inner loop of Algorithm 1 requires initializing a fresh optimizer for every outer loop iteration. Since we use

an outer-loop optimizer that also requires maintaining state (ScheduleFree AdamW (Defazio et al., 2024)), we move the outer loop optimizer to the CPU before instantiating inner-loop optimizers.

Second, first-order meta-learning in smaller models can typically be implemented by running multiple forward passes for each inner loop iteration, averaging losses, then backpropagating on the averaged loss term. However, because each inner-loop tamper-resistance loss term ($\mathcal{L}_{\text{TR}}$ in Algorithm 1) is computed on a separate forward pass, this requires maintaining $K$ computation graphs in memory. Since this is infeasible on reasonable hardware for LLMs with billions of parameters, we circumvent this inefficiency by accumulating tamper-resistance gradients in a separate data structure ($g_{\text{TR}}$ in Algorithm 1). We note that this can be done without using additional `all-gather` and `reduce-scatter` distributed operations, since tamper-resistance gradient accumulation and application to the pre-inner loop model parameters ($\theta_{i-1}$ in Algorithm 1) can be computed solely on sharded gradients.

## D ADDITIONAL ANALYSIS EXPERIMENTS

### D.1 BENIGN FINE-TUNING

|          |          | WMDP Forget ($\downarrow$) | Benign Domain ($\uparrow$) |
|----------|----------|------|------|
| TAR-Bio  | Pre-SFT  | 28.1 | 59.7 |
|          | Post-SFT | 30.1 | 64.7 |
| TAR-Chem | Pre-SFT  | 28.4 | 58.6 |
|          | Post-SFT | 27.5 | 62.8 |

Table 3: Average accuracy on MMLU economics subjects and WMDP Forget subjects for our Llama-3-8B TAR models safeguarded against hazardous biosecurity and chemical knowledge, before and after fine-tuning on benign economics data (Li et al., 2024). Results indicate that models safeguarded with TAR still preserve benign fine-tunability.

An important property of open-weight models is that they can be fine-tuned to improve performance on custom data or in specific domains. Thus, ideal tamper-resistant safeguards should allow continued fine-tuning of a model while preserving the safeguard. We evaluate whether TAR models can be fine-tuned on data unrelated to the safeguard using economics as an example domain. Using TAR models with biosecurity and chemical security safeguards, we perform supervised fine-tuning on the WMDP auxiliary economics corpora (Li et al., 2024). We fine-tune models for 2 epochs using a learning rate of $2 \times 10^{-6}$ and a batch size of 32, using AdamW ScheduleFree (Defazio et al., 2024). For evaluation, we report average accuracy across the corresponding MMLU subjects (High School Macroeconomics and Microeconomics) before and after fine-tuning. To confirm that the safeguard remains tamper-resistant in this setting, we also evaluate accuracy on corresponding WMDP subjects before and after fine-tuning.

We show the results of this evaluation in Table 3. We find that accuracy on economics questions can be improved by 5.0 percentage points without recovering significant hazardous knowledge. This illustrates that strong tamper-resistance can be compatible with benign model editing.

### D.2 TARGETED PATCHING OF VULNERABILITIES

In Section E.2, we discuss the set of adversaries sampled during TAR for each weaponization knowledge domain. However, we included the "Retain-set SFT followed by Weaponization-domain SFT" (R→F in Table 4) adversary to improve robustness to specific adversaries that broke the defense in earlier iterations of the method that did not sample this adversary. We find in Table 4 that including the R→F adversary significantly improves robustness to more test-time attacks, indicated by an improvement in Post-Attack Forget error by nearly 10.0 percentage points. Importantly, we find that including the 64-step R→F adversary at train-time improves test-time robustness to both 500-step R→F adversaries (Adv. 8 and 9) in Table 9 by nearly 30.0 percentage points on average, indicating that target patching of vulnerabilities is possible within TAR.

| Adversary Setup | Post-Attacks | |
|---|---|---|
| | All Advs. (↓) | R→F (↓) |
| Excl. R→F Adversary | 44.9 | 58.7 |
| Incl. R→F Adversary | **35.2** | **29.7** |

Table 4: Pre-Attack and Post-Attack scores when excluding and including the "Retain-set SFT followed by Biosecurity SFT adversary" (R→F) described in Appendix E.2. Post-attack R→F accuracy averaged over both Adv. 8 and 9 in Table 9. Average tamper-resistance across all 26 adversaries improves by nearly 10 percentage points when including the R→F adversary in TAR, and improves by nearly 30 percentage points for the R→F adversary specifically.

### D.3 TEST-TIME GENERALIZATION EXPERIMENTS

**Generalization to stronger test-time attacks.** In Figure 6, we show a biosecurity fine-tuning attack at an LR of $2 \times 10^{-5}$ on our TAR model and a model safeguarded with LLMU. We find that the tamper-resistance of TAR can generalize far beyond the $64$ steps used by train-time adversaries. Surprisingly, we observe that the test-time adversary's cross-entropy loss does not decrease below 7 for all $1,000$ steps. Moreover, the loss enters a plateau and does not decrease at all after 200 steps.

We also plot the entropy of the SFT adversary's posteriors during the attack (blue dashed line). For the first $\sim 200$ SFT attack steps, the adversary's entropy remains close to $\log(\texttt{vocab\_size})$ — the maximum possible entropy (shown as "Max Entropy" by the gray dashed line). This is expected, since this exactly what TAR optimizes for via the negative entropy tamper-resistance loss. The convergence to maximum entropy at test-time clearly demonstrates the TAR meta-learning objective working as intended, in line with the inner-loop posteriors observed in Figure 3.

As a point of reference, we show the progression of the same attack on LLMU. In this case, the adversary's cross-entropy loss decreases to within the recovery region in under 20 steps, corresponding to recovery in forget-set performance on WMDP.

**Varying the TAR inner-loop length $K$.** Recall that via the efficiency trick discussed in Appendix C.4, a single inner loop trajectory of length $K$ during TAR returns the $K$ sampled attacks in Algorithm 1. We compare the test-time loss robustness as we vary the length of the inner loop $K$ during TAR, running fine-tuning attacks for $1,000$ steps on a held-out forget dataset for biosecurity weaponization (Adversary 8 in Table 9).

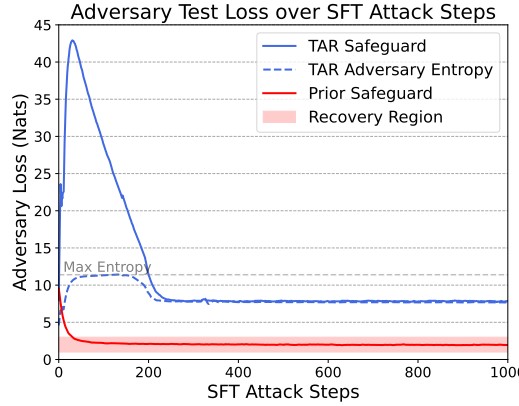

Figure 6: Our TAR safeguard can be robust to fine-tuning attacks that greatly exceed the $64$ steps used by train-time adversaries. For the LLMU safeguard, the adversary's loss quickly decreases into the recovery region. By contrast, TAR maintains flatness at a high loss for all $1,000$ steps. Solid lines are cross-entropy losses.

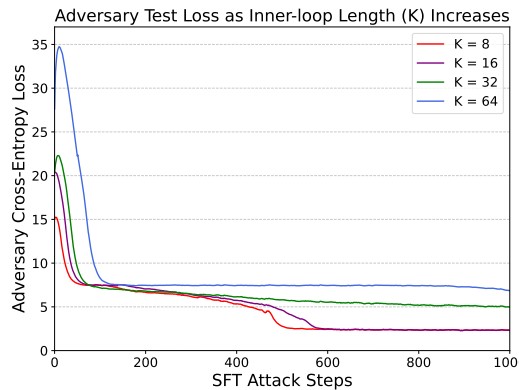

Figure 7: Comparison of a $1,000$-step SFT attack against TAR with the inner-loop length $K$ varied between $\{8, 16, 32, 64\}$. Test-time loss plateau magnitude and duration increase as $K$ increases.

For each value of $K$, we observe a plateau in the test loss that drops off at later steps as $K$ increases. This suggests that the robustness of TAR improves as the inner loop length increases. Prior work also corroborates that increasing the inner-loop length during meta-learning increases test-time generalization (Henderson et al., 2023). We note the contrast to conventional meta-learning methods mentioned in Section 4, in which typical meta-learning applications seek optimality after as few test-time steps as possible (Nichol et al., 2018; Finn et al., 2017). Here, our results suggest that the TAR objective is incentivized to run with as many inner loop steps as possible, representing a beneficial tradeoff in which compute can be exchanged for robustness. We find that $K = 64$ provides significant robustness to the range of adversaries we consider in Section 5 and Appendix F.1, while balancing computational efficiency as discussed in Appendix C.4.

## D.4 ABLATIONS

| Ablation | Pre-Attack | | Post-Attacks (Avg) |
|---|---|---|---|
| | Retain ($\uparrow$) | Forget ($\downarrow$) | Forget ($\downarrow$) |
| No Defense | 67.3 | 70.5 | 70.5 |
| Excl. Adv. Training | 59.7 | 27.3 | 61.6 |
| Excl. Initial Safeguard | 62.5 | 47.3 | 35.5 |
| TAR | 54.7 | **28.1** | **35.2** |

Table 5: Ablations for primary components of TAR: (1) the initial model safeguard, (2) the adversarial training phase. We find that these components are critical for the high tamper-resistance that TAR achieves.

**Including the initial safeguard.** In Table 5, we examine the impact of incorporating the *Random Mapping* safeguard step prior to the adversarial training phase during TAR. The *Random Mapping* safeguard in isolation achieves a near-random chance Pre-Attack Forget accuracy of 27.3 ("Excl. Adv. Training" in Table 5). However, it is susceptible to fine-tuning attacks similar to other baselines in Table 1, indicated by a higher Post-Attack Forget accuracy of 61.6. When including the tamper-resistance adversarial training phase (TAR), we observe significantly increased tamper-resistance as the Post-Attack Forget accuracy decreases by nearly 26 percentage points.

**Excluding the initial safeguard.** We also examine the impact of performing the adversarial training phase without the initial safeguarding step ("Excl. Initial Safeguard" in Table 5), finding that Pre-Attack accuracy is substantially higher without the initial safeguard. We find that including the *Random Mapping* phase reduces pre-attack forget set accuracy by 19.2 percentage points.

| $\mathcal{L}_{\text{TR}}$ Weighting | Pre-Attack | | Post-Attacks (Avg) |
|---|---|---|---|
| | Retain ($\uparrow$) | Forget ($\downarrow$) | Forget ($\downarrow$) |
| $\lambda_{\text{TR}} = 1.0$ | 62.5 | 29.3 | 39.9 |
| $\lambda_{\text{TR}} = 4.0$ | 54.7 | **28.1** | **35.2** |

Table 6: Pre-Attack and Post-Attack scores when varying the tamper-resistance loss weighting, $\lambda_{\text{TR}}$. Tamper-resistance improves by nearly 10.0% when increasing $\lambda_{\text{TR}}$ from 1.0 to 4.0. The retain loss weight $\lambda_{\text{retain}}$ is fixed at 1.0 for both settings.

**Varying the tamper-resistance loss scale $\lambda_{\text{TR}}$.** We compare the downstream robustness of TAR when varying the tamper-resistance loss weighting $\lambda_{\text{TR}}$ between 1.0 and 4.0 in Table 6. We observe that when setting $\lambda_{\text{TR}} = 1.0$, TAR maintains high retain MMLU accuracy at 62.5 percentage points, with moderate tamper-resistance indicated by a Post-Attack Forget accuracy of 39.9. Further increasing $\lambda_{\text{TR}}$ to 4.0 in our final TAR model results in a significantly improved Post-Attack Forget Accuracy of 35.2, with a partial decrease in Retain MMLU to 54.7. When varying $\lambda_{\text{TR}}$, we keep $\lambda_{\text{retain}}$ constant; thus, our results indicate a clear way to increase downstream tamper-resistance by increasing the weighting of the tamper-resistance gradient during TAR, reflecting a balance between tamper-resistance and capabilities similar to the robustness-performance tradeoff for adverarial robustness in vision models.

## D.5 DPO Tamper-Resistance during TAR

In Figure 8, we plot the DPO win-rate during harmful SFT attack trajectories during the adversarial training phase of TAR. We find that the outer-loop DPO loss steadily decreases, which corresponds to the average inner-loop win-rate of refusal completions over rejected completions steadily increasing over the 100 outer loop steps. Our results demonstrate that TAR is able to satisfy complex tamper-resistance losses after fine-tuning. We believe that this is a useful feature of the method, enabling TAR to adapt to other potentially useful objective functions that correspond to downstream robustness.

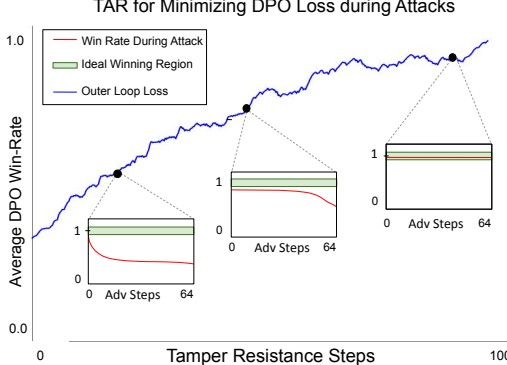

Figure 8: The development of inner-loop DPO win-rates during harmful SFT attack inner loops (red), over the course of tamper-resistance training for TAR. The outer loop win-rate (blue) depicts the average win-rate across inner loops over the course of tamper-resistance training. We observe that by the end of training, the win-rate for refusal completions becomes completely flat near the optimal win-rate value of 1.0.

## E EXPERIMENT DETAILS

### E.1 WEAPONIZATION DOMAIN PROXY DATASET DETAILS

**Biosecurity.** We use a synthetically labeled partition of the Pile (Gao et al., 2020) that was filtered for relevance to biology and the Camel AI Biology dataset (Li et al., 2023). We generate synthetic labels for Pile token sequences using openchat-3.5 (Wang et al., 2023), categorizing them as belonging to "Cellular Biology" or not. This process yields 49,984 samples: 7,558 for the Forget-set (Pile-bio Forget) and 42,426 for the Retain-set (Pile-bio Retain). Concurrently, we pack entries from the Camel AI Biology dataset to the truncation-enabled 256 tokenization limit, resulting in 54,258 samples of about 188 words each (Camel-bio Forget). We apply the same procedure to our held-out hazardous biology dataset (identical to the WMDP biosecurity Forget-set), producing 598,933 samples of similar length (OOD Forget).

**Chemical Security.** We use a private forget dataset containing text sequences about hazardous chemical security content (Chem Forget).

**Cybersecurity.** We scrape CTF writeups on CTFtime (CTFtime, 2024) that are numbered between 1 and 39181, collecting cybersecurity writeups written as recently as 2024. We filter to keep writeups that contain more than 150 characters. As a result of filtering and HTTP errors while scraping, our resulting forget dataset contains slightly over 18k samples (Cyber Forget).

### E.2 TRAIN-TIME SETTINGS AND ADVERSARIES

**Weaponization knowledge restriction.** For each weaponization knowledge restriction domain, we have a corresponding retain dataset $\mathcal{D}_{\text{retain}}$, comprised of a mix of data from the Pile-bio Retain set and Magpie-Align instruction-tuning dataset (Xu et al., 2024), which we refer to as the Retain-set. Referencing the datasets described in Appendix E.1, we specify the adversary tamper-resistance datasets that constitute the data used by attacks sampled from $\mathcal{A}_{\text{train}}$ during TAR training, as well as the attack setup for each domain as follows:

- **Biosecurity:** We simulate 6 adversaries from the following three setups: Pile-Bio Forget-set SFT, Camel-Bio Forget-set SFT, and Retain-set SFT followed by Pile-Bio Forget-set SFT (R→F), where the switching point between Retain-set and Pile-Bio Forget-set SFT within the 64-step length trajectory is sampled from a beta distribution $\beta(6.0, 3.0)$. LRs are sampled from $\{2 \times 10^{-5}, 4 \times 10^{-5}\}$.

- **Chemical Security:** We simulate 6 adversaries from the following three setups: Chem Forget-set SFT, Retain-set SFT, and Retain-set followed by Chem Forget-set SFT, using the same switching-

point sampling scheme as in the Biosecurity setting. LRs are sampled from $\{2 \times 10^{-5}, 4 \times 10^{-5}\}$. For chemical security, we set the tamper-resistance loss scale $\lambda_{\text{TR}}$ to 3.0.

- **Cybersecurity:** We simulate 6 adversaries from the following two setups: Cyber Forget-set SFT, Retain-set SFT, and Retain-set SFT followed by Cyber Forget-set SFT, using the same switching point sampling scheme as in the Chemical Security and Biosecurity settings. LRs are sampled from $\{2 \times 10^{-5}, 4 \times 10^{-5}\}$.

For each weaponization knowledge domain, we create 80-20 splits for adversary and held-out data of the corresponding forget sets, respectively. For Biosecurity, which uses multiple forget datasets, this involves creating 80-20 splits for each dataset, then combining the corresponding splits. The adversary data splits are used for sampled attacks from $\mathcal{A}_{\text{train}}$, whereas the held-out split is used for computing tamper-resistance losses. The held-out splits for each domain correspond to $\mathcal{D}_{\text{TR}}$ in Section 4. We use minibatches from a held-out dataset for computing tamper-resistance losses rather than cycling through a single dataset, following the recommendation of Nichol et al. (2018), in which first-order meta-learning without properly held-out minibatches caused a performance degradation.

All train-time adversary setups are tabulated in Table 7, where F-Pile, F-Chem, F-Cyber denote the respective datasets described in Appendix E.1, and Retain denotes the mixed Pile-bio and Magpie-Align Retain-set described in Appendix E.2. We use R→F to label the adversaries that perform Retain-set SFT followed by Forget-set SFT. The final column is an abbreviation for Finetuning Paradigm and indicates whether the SFT setup used full parameter finetuning or parameter-efficient fine-tuning (PEFT) via LoRA adapters (Hu et al., 2021).

Table 7: Train-time adversary setups for weaponization knowledge restriction of Biosecurity, Chemical Security, and Cybersecurity.

| Adversary | Dataset | Opt. Steps ($K$) | Optimizer | LR | LR Schedule | Batch Size | FT Paradigm |
|---|---|---|---|---|---|---|---|
| **Biosecurity Weaponization Restriction** | | | | | | | |
| Adv 1 | F-Pile | 64 | AdamW | $2 \times 10^{-5}$ | No Warmup | 64 | Full Parameter |
| Adv 2 | F-Pile | 64 | AdamW | $4 \times 10^{-5}$ | No Warmup | 64 | Full Parameter |
| Adv 3 | F-Camel | 64 | AdamW | $2 \times 10^{-5}$ | No Warmup | 64 | Full Parameter |
| Adv 4 | F-Camel | 64 | AdamW | $4 \times 10^{-5}$ | No Warmup | 64 | Full Parameter |
| Adv 5 | R→F | 64 | AdamW | $2 \times 10^{-5}$ | No Warmup | 64 | Full Parameter |
| Adv 6 | R→F | 64 | AdamW | $4 \times 10^{-5}$ | No Warmup | 64 | Full Parameter |
| **Chemical Security Weaponization Restriction** | | | | | | | |
| Adv 1 | F-Chem | 64 | AdamW | $2 \times 10^{-5}$ | No Warmup | 64 | Full Parameter |
| Adv 2 | F-Chem | 64 | AdamW | $4 \times 10^{-5}$ | No Warmup | 64 | Full Parameter |
| Adv 3 | Retain | 64 | AdamW | $2 \times 10^{-5}$ | No Warmup | 64 | Full Parameter |
| Adv 4 | Retain | 64 | AdamW | $4 \times 10^{-5}$ | No Warmup | 64 | Full Parameter |
| Adv 5 | R→F | 64 | AdamW | $2 \times 10^{-5}$ | No Warmup | 64 | Full Parameter |
| Adv 6 | R→F | 64 | AdamW | $4 \times 10^{-5}$ | No Warmup | 64 | Full Parameter |
| **Cybersecurity Weaponization Restriction** | | | | | | | |
| Adv 1 | F-Cyber | 64 | AdamW | $2 \times 10^{-5}$ | No Warmup | 64 | Full Parameter |
| Adv 2 | F-Cyber | 64 | AdamW | $4 \times 10^{-5}$ | No Warmup | 64 | Full Parameter |
| Adv 3 | Retain | 64 | AdamW | $2 \times 10^{-5}$ | No Warmup | 64 | Full Parameter |
| Adv 4 | Retain | 64 | AdamW | $4 \times 10^{-5}$ | No Warmup | 64 | Full Parameter |
| Adv 5 | R→F | 64 | AdamW | $2 \times 10^{-5}$ | No Warmup | 64 | Full Parameter |
| Adv 6 | R→F | 64 | AdamW | $4 \times 10^{-5}$ | No Warmup | 64 | Full Parameter |

**Harmful request refusal.** For harmful request refusal, we choose the retain dataset $\mathcal{D}_{\text{retain}}$ to be the Magpie-Align instruction-tuning dataset (Xu et al., 2024). We sample train-time adversaries that perform $K = 64$ steps of SFT on rejected completions of the Anthropic-HH-RLHF dataset (Bai et al., 2022) and vary the learning rate within $\{2 \times 10^{-6}, 2 \times 10^{-5}, 4 \times 10^{-5}\}$. We depict this list of adversaries in Table 8.

While the rejected completions from Anthropic-HH-RLHF constitute the data used for sampled attacks from $\mathcal{A}_{\text{train}}$ for harmful request refusal, we compute the tamper-resistance loss $\mathcal{L}_{\text{TR}}$ as follows. Since $\mathcal{L}_{\text{TR}}$ is the DPO loss (Rafailov et al., 2023) in this setting, we use the base model weights $\theta$ as the reference model and sample harmful and benign completions from a modified test split of Anthropic-HH-RLHF (Bai et al., 2022), where rejected completions are replaced with refusals (Cai et al., 2024). To avoid keeping the base model weights in memory and speed up training, we precompute the reference model DPO log-probabilities for the full Anthropic-HH-RLHF dataset before training. To summarize, we have that the sampled train-time adversaries perform SFT on the *rejected* completions from Anthropic-HH-RLHF, and tamper-resistance DPO loss is computed on the corresponding modified refusal completions; the modified refusal completions in this setting correspond to $\mathcal{D}_{\text{TR}}$ in Section 4.

In practice, we perform an additional 100 steps of supervised fine-tuning on the Magpie-Align dataset to improve the benign capabilities performance of the TAR refusal model in Table 2.

Table 8: Train-time adversary red-teaming setups harmful request refusal. "A-HH-Rejected" in the Dataset column corresponds to the adversary dataset comprised of rejected completions from the Anthropic-HH-RLHF dataset.

| Adversary | Dataset | Opt. Steps ($K$) | Optimizer | LR | LR Schedule | Batch Size | FT Paradigm |
|---|---|---|---|---|---|---|---|
| **Harmful Request Refusal** | | | | | | | |
| Adv 1 | A-HH-Rejected | 64 | AdamW | $2 \times 10^{-5}$ | No Warmup | 64 | Full Parameter |
| Adv 2 | A-HH-Rejected | 64 | AdamW | $2 \times 10^{-6}$ | No Warmup | 64 | Full Parameter |
| Adv 3 | A-HH-Rejected | 64 | AdamW | $4 \times 10^{-5}$ | No Warmup | 64 | Full Parameter |

## F    Red Teaming Details

### F.1    Weaponization Knowledge Restriction

In Table 9, we list all *test-time* adversary setups for recovering Biosecurity, Chemical Security, and Cybersecurity Weaponization knowledge.

For Biosecurity, we examine post-attack forget accuracy after fine-tuning for 500 steps on three data distributions: the Pile-bio Forget set and the Retain-set used during *Random Mapping* and TAR, and an OOD-Forget set mentioned in Appendix E.1, which is unseen during tamper-resistance training.

We leverage a consistent setup for Chemical Security and Cybersecurity, substituting the Pile-bio Forget set for the respective Chem and Cyber Forget sets. We exclude the unseen forget distribution adversaries for a total of 24 attacks, as we were unable to obtain an equally high quality held-out dataset for Chemical Security and Cybersecurity.

For all subjects, we compare the effect of different optimizers (AdamW, Adadelta, and Stochastic Gradient Descent with Nesterov Momentum, Schedule Free AdamW) (Zeiler, 2012; Kingma & Ba, 2017; Xie et al., 2023; Defazio et al., 2024), learning rates ($2 \times 10^{-6}, 2 \times 10^{-5}, 4 \times 10^{-5}$), and learning rate scheduling techniques (SGDR and 30 steps of linear warmup) (Loshchilov & Hutter, 2016).

Similar to Table 7, we use F-Pile, F-Chem, F-Cyber, and OOD-F in the Dataset Column to denote the respective datasets described in Appendices F.1. At test-time, we use the Pile-bio Retain set as the global Retain-set adversary, labeled as Retain. The R→F adversary at test-time also differs from the train-time version: we perform Forget-set SFT for 40% of the optimization steps, followed by Retain-set SFT for the remaining 60%. We found this combination to be a potent attack that broke intermediate versions of the method, as described in Appendix D.2.

In the Optimizer column, Schedule Free is an abbreviation of Schedule Free AdamW and SGD Nesterov is an abbreviation of SGD with Nesterov Momentum. In cases where the adversary used parameter-efficient fine-tuning (PEFT) via LoRA adapters (Hu et al., 2021), we used a LoRA config with an attention dimension, or rank, of 16, a LoRA alpha value of 32, a LoRA dropout of 0.05, on target linear modules:

{`up_proj`, `down_proj`, `gate_proj`, `q_proj`, `k_proj`, `v_proj`, `o_proj`}.

Lastly, for each Weaponization Knowledge Restriction domain, we red-team the TAR model with three runs of supervised fine-tuning attacks from each adversary, using different random seeds. We then calculate the final post-attack value for each adversary by averaging these three replicates.

Table 9: Test-time adversary red-teaming setups for weaponization knowledge restriction of Biosecurity, Chemical Security, and Cybersecurity.

| Adversary | Dataset | Opt. Steps ($K$) | Optimizer | LR | LR Schedule | Batch Size | FT Paradigm |
|---|---|---|---|---|---|---|---|
| **Biosecurity Weaponization Restriction** | | | | | | | |
| Adv 1 | F-Pile | 500 | AdamW | $2 \times 10^{-5}$ | No Warmup | 64 | Full Parameter |
| Adv 2 | F-Pile | 500 | AdamW | $4 \times 10^{-5}$ | No Warmup | 64 | Full Parameter |
| Adv 3 | Retain | 500 | AdamW | $2 \times 10^{-5}$ | No Warmup | 64 | Full Parameter |
| Adv 4 | Retain | 500 | AdamW | $4 \times 10^{-5}$ | No Warmup | 64 | Full Parameter |
| Adv 5 | OOD-F | 500 | AdamW | $2 \times 10^{-5}$ | No Warmup | 64 | Full Parameter |
| Adv 6 | OOD-F | 500 | AdamW | $4 \times 10^{-5}$ | No Warmup | 64 | Full Parameter |
| Adv 7 | R→F | 500 | AdamW | $2 \times 10^{-5}$ | No Warmup | 64 | Full Parameter |
| Adv 8 | R→F | 500 | AdamW | $4 \times 10^{-5}$ | No Warmup | 64 | Full Parameter |
| Adv 9 | F-Pile | 500 | Adadelta | $2 \times 10^{-5}$ | No Warmup | 64 | Full Parameter |
| Adv 10 | F-Pile | 500 | Adadelta | $4 \times 10^{-5}$ | No Warmup | 64 | Full Parameter |
| Adv 11 | F-Pile | 500 | Schedule Free | $2 \times 10^{-5}$ | No Warmup | 64 | Full Parameter |
| Adv 12 | F-Pile | 500 | Schedule Free | $4 \times 10^{-5}$ | No Warmup | 64 | Full Parameter |
| Adv 13 | F-Pile | 500 | SGD Nesterov | $2 \times 10^{-5}$ | No Warmup | 64 | Full Parameter |
| Adv 14 | F-Pile | 500 | SGD Nesterov | $4 \times 10^{-5}$ | No Warmup | 64 | Full Parameter |
| Adv 15 | F-Pile | 500 | AdamW | $2 \times 10^{-6}$ | No Warmup | 64 | Full Parameter |
| Adv 16 | F-Pile | 500 | AdamW | $2 \times 10^{-6}$ | 30 Steps Warmup | 64 | Full Parameter |
| Adv 17 | F-Pile | 500 | AdamW | $2 \times 10^{-5}$ | 30 Steps Warmup | 64 | Full Parameter |
| Adv 18 | F-Pile | 500 | AdamW | $4 \times 10^{-5}$ | 30 Steps Warmup | 64 | Full Parameter |
| Adv 19 | F-Pile | 500 | AdamW | $2 \times 10^{-5}$ | SGDR | 64 | Full Parameter |
| Adv 20 | F-Pile | 500 | AdamW | $4 \times 10^{-5}$ | SGDR | 64 | Full Parameter |
| Adv 21 | F-Pile | 500 | AdamW | $2 \times 10^{-5}$ | No Warmup | 32 | Full Parameter |
| Adv 22 | F-Pile | 500 | AdamW | $4 \times 10^{-5}$ | No Warmup | 32 | Full Parameter |
| Adv 23 | F-Pile | 500 | AdamW | $2 \times 10^{-5}$ | No Warmup | 128 | Full Parameter |
| Adv 24 | F-Pile | 500 | AdamW | $4 \times 10^{-5}$ | No Warmup | 128 | Full Parameter |
| Adv 25 | F-Pile | 500 | AdamW | $2 \times 10^{-5}$ | No Warmup | 64 | PEFT |
| Adv 26 | F-Pile | 500 | AdamW | $4 \times 10^{-5}$ | No Warmup | 64 | PEFT |
| **Chemical Security Weaponization** | | | | | | | |
| Adv 1 | F-Chem | 500 | AdamW | $2 \times 10^{-5}$ | No Warmup | 64 | Full Parameter |
| Adv 2 | F-Chem | 500 | AdamW | $4 \times 10^{-5}$ | No Warmup | 64 | Full Parameter |
| Adv 3 | Retain | 500 | AdamW | $2 \times 10^{-5}$ | No Warmup | 64 | Full Parameter |
| Adv 4 | Retain | 500 | AdamW | $4 \times 10^{-5}$ | No Warmup | 64 | Full Parameter |
| Adv 5 | R→F | 500 | AdamW | $2 \times 10^{-5}$ | No Warmup | 64 | Full Parameter |
| Adv 6 | R→F | 500 | AdamW | $4 \times 10^{-5}$ | No Warmup | 64 | Full Parameter |
| Adv 7 | F-Chem | 500 | Adadelta | $2 \times 10^{-5}$ | No Warmup | 64 | Full Parameter |
| Adv 8 | F-Chem | 500 | Adadelta | $4 \times 10^{-5}$ | No Warmup | 64 | Full Parameter |
| Adv 9 | F-Chem | 500 | ScheduleFree | $2 \times 10^{-5}$ | No Warmup | 64 | Full Parameter |
| Adv 10 | F-Chem | 500 | ScheduleFree | $4 \times 10^{-5}$ | No Warmup | 64 | Full Parameter |
| Adv 11 | F-Chem | 500 | SGD Nesterov | $2 \times 10^{-5}$ | No Warmup | 64 | Full Parameter |
| Adv 12 | F-Chem | 500 | SGD Nesterov | $4 \times 10^{-5}$ | No Warmup | 64 | Full Parameter |
| Adv 13 | F-Chem | 500 | AdamW | $2 \times 10^{-6}$ | No Warmup | 64 | Full Parameter |

**Table 9 – continued from previous page**

| Adversary | Dataset | Opt. Steps ($K$) | Optimizer | LR | LR Schedule | Batch Size | FT Paradigm |
|---|---|---|---|---|---|---|---|
| Adv 14 | F-Chem | 500 | AdamW | $2 \times 10^{-6}$ | 30 Steps Warmup | 64 | Full Parameter |
| Adv 15 | F-Chem | 500 | AdamW | $2 \times 10^{-5}$ | 30 Steps Warmup | 64 | Full Parameter |
| Adv 16 | F-Chem | 500 | AdamW | $4 \times 10^{-5}$ | 30 Steps Warmup | 64 | Full Parameter |
| Adv 17 | F-Chem | 500 | AdamW | $2 \times 10^{-5}$ | SGDR | 64 | Full Parameter |
| Adv 18 | F-Chem | 500 | AdamW | $4 \times 10^{-5}$ | SGDR | 64 | Full Parameter |
| Adv 19 | F-Chem | 500 | AdamW | $2 \times 10^{-5}$ | No Warmup | 32 | Full Parameter |
| Adv 20 | F-Chem | 500 | AdamW | $4 \times 10^{-5}$ | No Warmup | 32 | Full Parameter |
| Adv 21 | F-Chem | 500 | AdamW | $2 \times 10^{-5}$ | No Warmup | 128 | Full Parameter |
| Adv 22 | F-Chem | 500 | AdamW | $4 \times 10^{-5}$ | No Warmup | 128 | Full Parameter |
| Adv 23 | F-Chem | 500 | AdamW | $2 \times 10^{-5}$ | No Warmup | 64 | PEFT |
| Adv 24 | F-Chem | 500 | AdamW | $4 \times 10^{-5}$ | No Warmup | 64 | PEFT |
| **Cybersecurity Weaponization Restriction** | | | | | | | |
| Adv 1 | F-Cyber | 500 | AdamW | $2 \times 10^{-5}$ | No Warmup | 64 | Full Parameter |
| Adv 2 | F-Cyber | 500 | AdamW | $4 \times 10^{-5}$ | No Warmup | 64 | Full Parameter |
| Adv 3 | Retain | 500 | AdamW | $2 \times 10^{-5}$ | No Warmup | 64 | Full Parameter |
| Adv 4 | Retain | 500 | AdamW | $4 \times 10^{-5}$ | No Warmup | 64 | Full Parameter |
| Adv 5 | R→F | 500 | AdamW | $2 \times 10^{-5}$ | No Warmup | 64 | Full Parameter |
| Adv 6 | R→F | 500 | AdamW | $4 \times 10^{-5}$ | No Warmup | 64 | Full Parameter |
| Adv 7 | F-Cyber | 500 | Adadelta | $2 \times 10^{-5}$ | No Warmup | 64 | Full Parameter |
| Adv 8 | F-Cyber | 500 | Adadelta | $4 \times 10^{-5}$ | No Warmup | 64 | Full Parameter |
| Adv 9 | F-Cyber | 500 | ScheduleFree | $2 \times 10^{-5}$ | No Warmup | 64 | Full Parameter |
| Adv 10 | F-Cyber | 500 | ScheduleFree | $4 \times 10^{-5}$ | No Warmup | 64 | Full Parameter |
| Adv 11 | F-Cyber | 500 | SGD Nesterov | $2 \times 10^{-5}$ | No Warmup | 64 | Full Parameter |
| Adv 12 | F-Cyber | 500 | SGD Nesterov | $4 \times 10^{-5}$ | No Warmup | 64 | Full Parameter |
| Adv 13 | F-Cyber | 500 | AdamW | $2 \times 10^{-6}$ | No Warmup | 64 | Full Parameter |
| Adv 14 | F-Cyber | 500 | AdamW | $2 \times 10^{-6}$ | 30 Steps Warmup | 64 | Full Parameter |
| Adv 15 | F-Cyber | 500 | AdamW | $2 \times 10^{-5}$ | 30 Steps Warmup | 64 | Full Parameter |
| Adv 16 | F-Cyber | 500 | AdamW | $4 \times 10^{-5}$ | 30 Steps Warmup | 64 | Full Parameter |
| Adv 17 | F-Cyber | 500 | AdamW | $2 \times 10^{-5}$ | SGDR | 64 | Full Parameter |
| Adv 18 | F-Cyber | 500 | AdamW | $4 \times 10^{-5}$ | SGDR | 64 | Full Parameter |
| Adv 19 | F-Cyber | 500 | AdamW | $2 \times 10^{-5}$ | No Warmup | 32 | Full Parameter |
| Adv 20 | F-Cyber | 500 | AdamW | $4 \times 10^{-5}$ | No Warmup | 32 | Full Parameter |
| Adv 21 | F-Cyber | 500 | AdamW | $2 \times 10^{-5}$ | No Warmup | 128 | Full Parameter |
| Adv 22 | F-Cyber | 500 | AdamW | $4 \times 10^{-5}$ | No Warmup | 128 | Full Parameter |
| Adv 23 | F-Cyber | 500 | AdamW | $2 \times 10^{-5}$ | No Warmup | 64 | PEFT |
| Adv 24 | F-Cyber | 500 | AdamW | $4 \times 10^{-5}$ | No Warmup | 64 | PEFT |

## F.2 HARMFUL REQUEST REFUSAL

For the harmful request refusal setting, we conduct 5 test-time adversary attacks that perform SFT for 10 epochs on a held-out toxicity dataset called Toxic-DPO v0.2, on each of the settings in Table 10. The dataset contains 541 user-assistant chat interactions where the assistant complies with harmful instructions.

Table 10: Test-time adversary red-teaming setups for harmful request refusal. "ToxicDPO" in the Dataset column refers to the ToxicDPOv0.2 dataset containing harmful chat completions.

| Adversary | Dataset | Epochs | Optimizer | LR | LR Schedule | Batch Size | FT Paradigm |
|---|---|---|---|---|---|---|---|
| **Harmful Request Refusal** | | | | | | | |
| Adv 1 | ToxicDPO | 10 | AdamW | $1 \times 10^{-5}$ | 10 Steps Warmup | 32 | Full Parameter |

**Table 10 – continued from previous page**

| Adversary | Dataset | Epochs | Optimizer | LR | LR Schedule | Batch Size | FT Paradigm |
|-----------|---------|--------|-----------|-----|-------------|------------|-------------|
| Adv 2 | ToxicDPO | 10 | AdamW | $1 \times 10^{-5}$ | No Warmup | 32 | Full Parameter |
| Adv 3 | ToxicDPO | 10 | AdamW | $1 \times 10^{-5}$ | No Warmup | 16 | Full Parameter |
| Adv 4 | ToxicDPO | 10 | AdamW | $2 \times 10^{-5}$ | No Warmup | 32 | Full Parameter |
| Adv 5 | ToxicDPO | 10 | AdamW | $4 \times 10^{-5}$ | No Warmup | 32 | Full Parameter |

### F.2.1 ADDITIONAL HARMFUL REQUEST REFUSAL RESULTS

| Model | Pre-Attacks | | Post-Attacks (Avg) |
|-------|-------------|-----|--------------------|
| | MT-Bench ($\uparrow$) | ASR ($\downarrow$) | ASR ($\downarrow$) |
| Refusal Trained | 8.1 | 14.7 | 72.5 |
| R2D2 | 6.0 | 25.0 | 78.3 |
| RepNoise | 6.2 | 18.8 | 74.5 |
| RR | 8.0 | **1.4** | 84.8 |
| TAR (Ours) | 6.3 | 31.4 | **63.9** |

Table 11: Average Post-Attack HarmBench ASR, reported for TAR, Representation Rerouting (RR), and the Refusal Trained Llama-3-8B-Instruct model across 5 fine-tuning attacks depicted in Table 10, as well as Pre-Attack MT-Bench and HarmBench ASR. TAR is more robust than other methods after tampering, while maintaining comparable MT-Bench performance. Note that Pre-Attack ASR is not a priority for us, as we focus on reducing ASR after tampering attacks. To improve both metrics, future work could consider combining tamper-resistance training with a strong baseline safeguard like RR. ASR values are percentages.

## G   BASELINE DETAILS

### G.1   WEAPONIZATION KNOWLEDGE RESTRICTION

**Max Entropy.**   Let $K$ be the set of all token-wise output probability distributions returned by a model $\theta$, where $k \in K$ corresponds to every position in the sequence. We maximize the average entropy of these discrete distributions in $K$ as follows:

$$\mathcal{L}_{\text{Max Entropy}} = \sum_{k \in \mathcal{K}} p_k \log(p_k)$$

This is equivalent to minimizing the average Kullback-Leibler (KL) divergence (Kullback & Leibler, 1951) between each $k$ and the discrete uniform distribution $u(x)$ over the vocabulary $V$. For Llama-3-8B-Instruct, $|V| = 128256$. Thus, this objective is upper-bounded by:

$$h(x) = -\log(u(x)) = \log(|V|) \approx 11.76$$

where $h(x)$ measures the Shannon information or self-information and $\log(x)$ has base $e$. We apply this objective for all elements in the Forget-set and perform standard cross-entropy on the Retain-set.

**Min Posterior.**   The goal of the Min Posterior objective is to assign lower probabilities to true forget-set labels, essentially minimizing $-\log(1 - P(\text{label}))$. Let $p_i$ be the probability assigned to the true label for token $i$ and $\mathcal{V}$ be the model's vocabulary distribution. We define the Min Posterior objective function as follows:

$$\mathcal{L}_{\text{Min Posterior}} = -\frac{1}{|\mathcal{V}|} \sum_{i \in \mathcal{V}} \log(1 - p_i + \epsilon) \cdot \mathbb{I}[\log(p_i) \geq \tau]$$

where $\tau$ is the threshold for masking out target label logits (which we set to the negative maximum entropy of the vocabulary distribution, $-\log |\mathcal{V}|$) and $\mathbb{I}[\cdot]$ is the corresponding indicator function (1 if the condition is true, 0 otherwise). We include an optional $\epsilon = 1 \times 10^{-12}$ to help with numerical stability. Similar to the Max Entropy objective, we apply this objective for all elements in the Forget-set and perform standard cross-entropy on the Retain-set.

**RMU.** We adapt RMU's implementation from Li et al. (2024) with a learning rate of $5 \times 10^{-5}$ and 250 unlearning steps. We use the released WMDP's unlearning datasets for Biosecurity (Bio) and Cybersecurity (Cyber) unlearning, and our private hazardous chemistry dataset for Chemical Security (Chem) unlearning. We use unlearning coefficients of 20, 30, and 50 for Bio, Cyber, and Chem respectively. We use a retain coefficient of 700 on Wikitext Merity et al. (2016).

**LLMU.** We use a modified version of LLMU from Yao et al. (2023). Instead of computing the KL divergence to regularize retain-set logits towards the base frozen model, we employ a standard cross-entropy loss. This modification allows for memory-efficient execution on our hardware while maintaining comparable performance.

**Hyperparameter tuning.** Besides RMU, all baseline hyperparameters were chosen after a grid search across learning rates $\{3 \times 10^{-6}, 5 \times 10^{-6}, 8 \times 10^{-6}, 1 \times 10^{-5}\}$, optimization step count $\{600, 1000\}$, and warmup steps $\{0, 100\}$. We found that 600 optimization steps using the AdamW Schedule Free optimizer, at a learning rate of $1 \times 10^{-5}$, with 100 steps of linear warmup, and an effective batch size of 64 produced the best performance. For the Max Entropy, Min Posterior, and LLMU baselines, we train on the three corresponding forget datasets discussed in E.1. For these baselines, we modify our Biosecurity forget corpus to be a mixture of the Pile-bio and Camel-bio Forget corpora. We use the Pile-bio Retain-set as a global Retain-set for baseline training.

## G.2 Harmful Request Refusal

**Representation Rerouting.** We use the Llama-3-8B-Instruct RR model from Zou et al. (2024), which uses a cosine distance loss to push representations for harmful inputs to become orthogonal to those of the base Llama-3-8B-Instruct model.

**R2D2.** We use the R2D2 model run on Zephyr-7B directly from Mazeika et al. (2024), which performs adversarial training against GCG attacks to increase jailbreak robustness.

**RepNoise.** We the RepNoise model run on Llama-2-7B directly from Rosati et al. (2024b), which uses a distributional loss to push representations for harmful inputs toward Gaussian noise.

## G.3 Additional Baseline Comparisons

**MLAC-AR.** Meta-Learned Adversarial Censoring (MLAC) (Henderson et al., 2023) was originally proposed to prevent BERT-style models from learning binary classification for gender bias data. Since the approach is not immediately applicable to LLMs, we extend MLAC in a variant we call autoregressive MLAC (MLAC-AR). Since MLAC in its original formulation calls for "task-blocking" via negating the adversary's loss during the inner loop of meta-learning, we implement this by negating the cross-entropy loss of an LLM fine-tuning adversary. However, we found that this approach diverges in performance across a variety of hyperparameters, and opted to further improve performance of the MLAC-AR baseline by clamping the maximum cross-entropy loss at the value of the maximum entropy of the output vocabulary distribution, `log(vocab_size)`. We show results in Table 12, finding that MLAC-AR does not maintain sufficient benign capabilities performance nor uniform tamper-resistance across weaponization domains.

**SOPHON-AR.** In concurrent work, SOPHON (Deng et al., 2024) was introduced to prevent small diffusion models and image classifiers from learning specific data distributions. Similarly to MLAC-AR, we extend SOPHON to LLMs via SOPHON-AR, using the alternating retain loss and fine-tuning suppression loss formulation that the authors propose. Furthermore, we adapt the inverse cross-entropy loss from Deng et al. (2024), which aims to boost convergence of the fine-tuning

| Domain | Model | Pre-Attacks | | Post-Attacks (Avg) |
|---|---|---|---|---|
| | | Retain ($\uparrow$) | Forget ($\downarrow$) | Forget ($\downarrow$) |
| Biosecurity | Random | 25.0 | 25.0 | 25.0 |
| | MLAC-AR | 49.1 | 31.2 | 50.6 |
| | SOPHON-AR | 27.2 | 24.0 | 28.3 |
| | TAR (Ours) | **54.7** | 28.1 | 35.2 |
| Chemical Security | Random | 25.0 | 25.0 | 25.0 |
| | MLAC-AR | 47.8 | 29.9 | 33.6 |
| | SOPHON-AR | 23.3 | 26.2 | 26.1 |
| | TAR (Ours) | **56.5** | 28.4 | 27.1 |
| Cybersecurity | Random | 25.0 | 25.0 | 25.0 |
| | MLAC-AR | 36.0 | 26.6 | 35.1 |
| | SOPHON-AR | 24.4 | 24.6 | 30.4 |
| | TAR (Ours) | **60.7** | 23.6 | 28.6 |

Table 12: Additional baselines for MLAC-AR, an extension of the method in Henderson et al. (2023) to autoregressive LLMs, as well as SOPHON-AR from Deng et al. (2024), respectively. Despite extensive tuning, SOPHON-AR does not yield a usable model. Additionally, MLAC-AR has varying robustness and worse Retain MMLU performance.

suppression process. We find in practice that despite heavy tuning, SOPHON-AR does not converge well enough to yield a usable Pre-Attack model in Table 12.

