# OpenReview forum: "Tamper-Resistant Safeguards for Open-Weight LLMs"
_ICLR.cc/2025/Conference — ICLR 2025 Poster_

### Official Review · Reviewer_i3oQ · 2024-10-26

**Soundness:** 3
**Presentation:** 2
**Contribution:** 2
**Rating:** 5
**Confidence:** 3

**Summary:**

The paper proposes Tampering Attack Resistance (TAR) method which builds robust safe LLMs under weight tampering attacks. The method achieves superior performance on weaponization knowledge restriction and harmful refusal training.

**Strengths:**

1. The defense of LLM's weight tampering attacks is an important topic. This work has great significance.

**Weaknesses:**

1. there are to many proxy objectives, like "safety_metric", "capabilities_metric". These pre-defined metrics will limit the significance and universality of the proposed method. Unless the author can prove that TAR is still working under other "safety_metric" or "capabilities_metric".
2. From Eq.1, TAR is a simple adversarial training paradigm, with some proxy indicators. While adversarial training is an exist well known technique.

**Questions:**

1. Are there any experiments to validate TAR on proxy metrics has a certain degree of generalization on other safety scenes or benchmarks?
2. How about the training cost of TAR. Is it larger than original training?
3. Except for the adversarial training, are there any novel findings or modifications of TAR, which suggest the novelty.

---

> ### Author Response · Authors · 2024-11-19
> **Author Response (1/2)**
>
> Thank you for your careful analysis of our work. We are glad you found our work to be of great significance, and we hope the following response addresses your concerns.
>
> **Definition of metrics.**
>
> > There are to many proxy objectives, like "safety_metric", "capabilities_metric". These pre-defined metrics will limit the significance and universality of the proposed method. Unless the author can prove that TAR is still working under other "safety_metric" or "capabilities_metric"
>
> In our method description, we only have two proxy objectives: one for safety and one for capabilities. These are the training losses that we use to improve the test-time safety and capabilities metrics. In other words, each application of our method only needs to consider two losses and two test-time metrics, which is fairly standard for the field.
>
> > Are there any experiments to validate TAR on proxy metrics has a certain degree of generalization on other safety scenes or benchmarks?
>
> Yes, we do have these experiments! Each of the two domains that we describe in section 3.2 uses different specific safety metrics, so we already have results demonstrating that TAR works under multiple different safety and capabilities metrics. We will clarify these points in the updated paper. Thank you for your feedback.
>
>
> **TAR training cost.**
>
> > How about the training cost of TAR. Is it larger than original training?
>
> Each TAR fine-tuning run required 18 hours on 8 NVIDIA A100 80GB GPUs. This is approximately 8 orders of magnitude less compute than original pre-training for the Llama-3-8B model. For details, please see below.
>
> **Regarding run-time overhead**, for each inner loop step, both an adversary gradient and meta-gradient is computed for the adversary’s loss and tamper-resistance loss, respectively. The outer loop also computes one additional gradient for the retain-set, which is added to the accumulated meta-gradient. However, via the subsampling trick discussed in Appendix B.3, we compute meta-gradients every 4 adversary steps, which we found to speed up training by a constant factor without sacrificing robustness. In total, there are (750 retain gradients + 750\*64 adversary gradients + 750\*64/4 meta-gradients) = 60,750 forward/backward passes computed during our method with a batch size of 64.
>
> This is multiple orders of magnitude fewer forward/backward passes than used in original model pre-training, which is reported to be 1,200,000 optimization steps with a batch size of 8,000,000 for Llama-3-8B [1]. This comes out to a factor of around 10^8 fewer forward/backward passes for our TAR fine-tuning method compared to Llama-3-8B pre-training.
>
> **Regarding memory overhead**, one might consider memory to be the primary concern for running TAR on large models. However, since we perform first-order meta-learning, as we explain in Appendix B.4, meta-gradients can be accumulated in-place in a separate data structure. Even with common model sharding algorithms such as FSDP, this can be performed without additional distributed communication, requiring additional memory equivalent to **only one extra copy of model gradients per device.** Thus, our method is quite memory-efficient as it scales memory usage by only a constant factor with sharded model size, which means **TAR has the same memory complexity as standard training asymptotically.**
>
> Thank you for bringing this up, we will include more discussion of these costs and our mitigations in our revision.
>
> [1] Llama Team, AI @ Meta. The llama 3 herd of models, 2024.

---

> > ### Author Response · Authors · 2024-11-19
> > **Author Response (2/2)**
> >
> > **Adversarial training formulation and method novelty.**
> >
> > > From Eq.1, TAR is a simple adversarial training paradigm, with some proxy indicators. While adversarial training is an exist well known technique.
> >
> > We apologize for any lack of clarity in our formulation. Please note that in Eq. 1, the `attack` applied to the model parameters $\theta$, *modifies the model parameters*. In traditional adversarial training, attacks are conducted on *model inputs*. Thus, our objective, while visually similar, is distinctly a meta-learning objective, since computing a gradient on our objective directly involves back-propagation through an inner-loop of fine-tuning.
> >
> > We wanted to draw a connection between our objective and adversarial training to bridge the gap for readers, since we do consider our method to be a *form* of adversarial training. However, we agree that this may introduce some confusion to the readers, as one might mistakenly assume that existing results are applicable to our setting. Thank you for pointing this out, we will improve our portrayal of Eq. 1 in the revision.
> >
> > > Except for the adversarial training, are there any novel findings or modifications of TAR, which suggest the novelty.
> >
> > As pointed out above, the objective of our method is distinct from standard adversarial training, as it involves attacks that tamper with model weights. Furthermore, our main contribution is to show that tamper-resistance for LLMs is tractable in the first place, as the method we introduce is the first to significantly raise the floor for tamper-resistance, since prior work is unsuccessful at defending nearly all of the attacks we consider. This itself is a significant technical contribution that we highlight throughout the paper.
> >
> > Additionally, our approach to solving this objective via TAR involves numerous technical and experimental contributions, including:
> >
> > - Developing a tamper-resistance evaluation pipeline for LLMs, including an extensive red teaming suite. This did not exist prior to our work.
> > - Demonstrating that MLAC does not scale to LLMs
> > - Developing a 2-stage method with an initial unlearning/alignment stage that doesn’t use tamper-resistance training, which greatly outperforms prior work
> > - Demonstrating that input-based adversarial training (e.g., R2D2) does not generalize to provide robustness to weight tampering attacks. This had not previously been shown.
> > - Demonstrating that concurrent work (e.g., RepNoise) lacks robustness against a broader set of attacks
> > - Identifying a significantly improved outer loop loss via maximizing entropy for tamper-resistance training
> > - Introduced methods for minimizing memory overhead during meta-learning, making our adversarial meta-learning procedure possible.
> >
> > Having made promising early progress on tamper-resistance, which was previously considered impossible, we believe these contributions would be of interest to the research community and could foster useful discussion at ICLR.
> >
> > If we have addressed the thrust of your concerns, we kindly ask that you consider raising your score.

---

> ### Comment · Reviewer_i3oQ · 2024-11-25
> **Thanks for your response.**
>
> Thanks the authors for the comprehensive response. The response addressed some of my concerns. However, for the core idea, although authors propose their method compute adversarial gradients on model parameters instead of model inputs. I still find there are previous works [1] have consider the adversarial weight perturbation. The core contribution of this paper is not convincing for me. Considering the authors make a comprehensive study on Weight Tampering Attack, I have decided to keep my original rating.
>
> [1] Adversarial Weight Perturbation Helps Robust Generalization. Dongxian Wu, Shu-tao Xia, Yisen Wang

---

> > ### Author Response · Authors · 2024-11-25
> >
> > Thank you for your response, we sincerely appreciate your engagement during the rebuttal period. We are glad you found our initial response addressed some of your concerns. However, we strongly believe that there is a misunderstanding regarding the novelty of our contribution, which we clarify below.
> >
> > We fully agree that the basic idea of adversarial perturbations on parameters has been explored before. In fact, our related work section contains several examples of this. The adversarial weight perturbation paper is another relevant prior work, and we will add it to this section thanks to your suggestion, although the problem we focus on is very different.
> >
> > Our main contribution is not to introduce the basic idea of adversarial perturbations on parameters. Rather, our contribution is developing a specific method based on this idea that scales up to LLMs and greatly improves robustness to parameter-based attacks. No prior work has demonstrated this before. Relatedly, we would like to highlight that Reviewer NvnF mentions our results to be “comprehensive and convincing”, and that Reviewer n8EQ finds our method to constitute a “substantial novel contribution to the field.”
> >
> > In our experiments, we compare to several prior and concurrent works that have explored adversarial perturbations on parameters. For example, we show that MLAC does not scale well to LLMs, while our method is the first to obtain substantial tamper-resistance.
> >
> > Regarding the Adversarial Weight Perturbation paper in particular, please note that they consider the threat model of input-based attacks and view weight perturbations as another way to improve robustness to standard input-based attacks. By contrast, the threat model we consider is one where the test-time adversary mounts parameter-based attacks, which is a far more challenging threat model that researchers have only begun to study relatively recently. To help clarify this point to readers, we will discuss these differences in the updated paper. Thank you for your suggestion. If we have addressed your remaining concerns, we kindly ask that you consider raising your score.

---

> > > ### Author Response · Authors · 2024-12-02
> > > **Friendly reminder**
> > >
> > > We would like to gently remind the reviewer that the rebuttal discussion period is ending soon, and that we are happy to answer any remaining questions or concerns.

---

### Official Review · Reviewer_NvnF · 2024-11-03

**Soundness:** 3
**Presentation:** 3
**Contribution:** 3
**Rating:** 6
**Confidence:** 4

**Summary:**

This paper studies an important problem: fine-tuning attacks on LLMs. They propose a novel defense method, TAR, to improve the robustness of LLMs against possible malicious fine-tuning attacks. This method is based on adversarial training and meta-learning, and a novel training objective combining both an adversarial objective and a retaining objective is proposed to maintain utility. Extensive experiments are conducted to illustrate the effectiveness of proposed method.

**Strengths:**

This paper is written well and logically. I enjoy reading this work and every detail is properly described. The large-scale experiments are comprehensive and convincing.

**Weaknesses:**

1. The time complexity analysis is important but not included. Both adversarial training and meta-learning are time-consuming. The proposed method can be expensive especially when the model size is increasing. This brings a concern about whether this method is practical for protecting large models. I suggest the authors provide computation analysis either empirical or theoretical.

2. There are many hyperparameters in either objective in Eq (1) or optimizing it, such as the number of outer loops and coefficients before tamper-resistance loss and retain loss (lambda_TR and lambda_retain). How they influence the defending performance is not discussed, and I suggest including it.

**Questions:**

I am curious about what kind of A_train in the objective in Eq(1) is required to have a good defense. For example, how diverse attacks need to be included; how many adversarial examples for each attack need to be included, etc.

---

> ### Author Response · Authors · 2024-11-19
> **Author Response (1/2)**
>
> Thank you for your careful analysis of our work. We are glad that you found our paper to be well-written, and our experiments to be comprehensive!  We hope the following response addresses your concerns/questions
>
> **Time and memory complexity of TAR / our efficiency optimizations.**
>
> > The time complexity analysis is important but not included … I suggest the authors provide computation analysis either empirical or theoretical.
>
> Your concern about computational efficiency for large models is valid, as sampling fine-tuning attacks and performing meta-learning naively may increase both runtime and memory overhead. However, we address both of these concerns in our paper, introducing several efficiency tricks in Appendix B.3 and B.4  to make first-order meta-learning possible and efficient for billion-parameter LLMs. We provide additional analysis for each of your concerns below for understanding the costs and how we mitigate them, as discussed in our paper.
>
> > Both adversarial training and meta-learning are time-consuming. The proposed method can be expensive especially when the model size is increasing.
>
> Our method, TAR, involves an inner-loop/outer-loop training setup, for conducting first-order meta-learning as described in both Algorithm 1 and Equation 1. For outer loop steps $N$ and inner loop steps $K$, the runtime complexity of the method is $O(NK)$. Generally, $K<<N$, as we observe empirically in Figure 6 in our appendix that inner-loop lengths of $K=64$ with an outer loop length of $N=750$, can already preserve a high adversary loss for much more than $K$ steps.
>
> Regarding run-time overhead, for each inner loop step, both an adversary gradient and meta-gradient is computed for the adversary’s loss and tamper-resistance loss, respectively. The outer loop also computes one additional gradient for the retain-set, which is added to the accumulated meta-gradient. However, via the subsampling trick discussed in Appendix B.3, we compute meta-gradients every 4 adversary steps, which we found to speed up training by a constant factor without sacrificing robustness. In total, there are (750 retain gradients + 750\*64 adversary gradients + 750\*64/4 meta-gradients) = 60,750 forward/backward passes computed during our method with a batch size of 64.
>
> This is multiple orders of magnitude fewer forward/backward passes than used in original model pre-training, which is reported to be 1,200,000 forward/backward passes with a batch size of 8,000,000 for Llama-3-8B [1].
>
> **On 8 80GB A100 GPUs, our method with N=750 and K=64 can be run in 18 hours.**
>
> > The proposed method can be expensive especially when the model size is increasing. This brings a concern about whether this method is practical for protecting large models.
>
> Regarding memory overhead, one might consider memory to be the primary concern for running TAR on large models. However, since we perform first-order meta-learning, as we explain in Appendix B.4, meta-gradients can be accumulated in-place in a separate data structure. Even with common model sharding algorithms such as FSDP, this can be performed without additional distributed communication, requiring additional memory equivalent to **only one extra copy of model gradients per device.** Thus, our method is quite memory-efficient as it scales memory usage by only a constant factor with sharded model size, which means **TAR has the same memory complexity as standard training asymptotically.**
>
> Thank you for bringing up this concern, we will include more discussion of these costs and our mitigations in our revision.
>
> [1] Llama Team, AI @ Meta. The llama 3 herd of models, 2024.

---

> ### Author Response · Authors · 2024-11-19
> **Author Response (2/2)**
>
> **Understanding the hyperparameters of TAR.**
>
> > There are many hyperparameters in either objective in Eq (1) or optimizing it, such as the number of outer loops and coefficients before tamper-resistance loss and retain loss (lambda_TR and lambda_retain). How they influence the defending performance is not discussed, and I suggest including it.
>
> We agree that discussing the effect of hyperparameters is important. In fact, we do include detailed discussion of important hyperparameters in our ablations in Appendix C.2 and C.4. In Figure 6 of Appendix C.2, we show the effect of increasing the inner loop steps $K$. In Table 6 of Appendix C.4, we actually already discuss the effect of varying $\lambda_{TR}$ as you suggest. We found that increasing $\lambda_{TR}$ from $1.0$ to $4.0$, while holding $\lambda_{\text{retain}}$ at $1.0$, greatly improved overall tamper-resistance, in line with our expectations.
>
> Regarding the outer loop steps $N$, we ran our method until we observed convergence in our full TAR loss (Eqn. 1). The smoothed outer-loop TAR loss is shown as the blue curves in Figure 3. We mention in Section 5.1 that $N=750$ for the weaponization knowledge restriction setting, and in Section 5.2 that $N=100$ for harmful request refusal.
>
> To clarify this for readers, we will add a list of all hyperparameters to the updated paper in one location for ease of reference, with recommended settings from our ablations. Thank you for your suggestion.
>
> **Sampling attacks during TAR.**
>
> > I am curious about what kind of A_train in the objective in Eq(1) is required to have a good defense. For example, how diverse attacks need to be included; how many adversarial examples for each attack need to be included, etc.
>
> This is a great question! We generally found that diversity in both dataset distribution and learning rate contribute significantly to robustness. For example, in Section 3.3, we mention that while developing TAR, we included new fine-tuning attacks that broke intermediate versions of the method. In one interesting case, we were concerned about an adversary that initially performs benign, retain-set fine-tuning, followed by forget-set fine-tuning (called the “R->F adversary”). We show in Table 4 in Appendix C.3 that targeted patching of vulnerabilities is possible with TAR, as sampling a reduced version of this R->F adversary actually did improve downstream robustness. This demonstrates that increasing the diversity of training attacks can improve downstream robustness.
>
> Overall, the configuration of training adversaries listed in Table 7 and 8 obtained the best robustness in our experiments for the domains we consider, which we hope future work will improve upon.
>
> If we have addressed the thrust of your concerns, we kindly ask that you consider raising your score.

---

> > ### Comment · Reviewer_NvnF · 2024-11-25
> >
> > Thank the authors for the complete response. I would like to keep the score.

---

### Official Review · Reviewer_vbKt · 2024-11-03

**Soundness:** 2
**Presentation:** 2
**Contribution:** 2
**Rating:** 5
**Confidence:** 3

**Summary:**

This paper introduces a novel method called TAR, designed to enhance the robustness of large language models (LLMs) against tampering attacks, addressing significant vulnerabilities in existing safeguards.

**Strengths:**

1. The topic studied in this paper is of great significance. Malicious LLMs can cause serious harm, such as spreading false news about public figures, causing discrimination and unfair results due to prejudice, and generating violent terrorist information. Therefore, it is necessary to apply robust safeguards to open source LLMs.

2. The proposed method successfully resists thousands of malicious fine-tuning.

**Weaknesses:**

1. The organization of Section 4 requires further refinement for improved readability. It would be beneficial to briefly outline the design motivation before delving into specific details, particularly regarding the content of Fig. 3. Additionally, the mathematical formulation of the loss function used is currently absent.

2. The caption of figure 1 needs to explain the difference between the two branches more clearly. For example, what's the difference between the first nodes of the two branches.

3. In Section 1, the authors assert that the proposed method can endure fine-tuning of up to 5000 steps. However, this claim does not intuitively convey the contribution of the paper. Firstly, the details surrounding fine-tuning, such as batch size and learning rate, are unclear; these parameters significantly influence the number of fine-tuning steps. Secondly, a comparative analysis with the typical number of steps required to fine-tune models on downstream tasks is lacking.

4. The threat model lacks clarity. The authors assume that the attacker is compute-bounded but do not provide a clear definition of what this entails. Furthermore, including concrete examples of the metrics for capabilities_metric and safety_metric would enhance understanding.

5. In section 4.2, the authors emphasized that the proposed method is different from standard meta-learning. However,  the differences highlighted seem minor and do not present significant technical challenges.

6. The term 'empirically' is employed multiple times in the methods section. While drawing conclusions from empirical observations is valuable for informing solution design, relying solely on empirical data, particularly in the selection of a loss function, may impose limitations on the solution's robustness. A theoretical analysis comparing the efficacy of the entropy loss function versus cross-entropy loss is necessary.

7. The performance metrics used in the experiments require clear explanation. Based on the results presented, the proposed solution appears to sacrifice task performance in favor of enhanced robustness, suggesting potential flaws in the method.

**Questions:**

None

**Details Of Ethics Concerns:**

N.A.

---

> ### Author Response · Authors · 2024-11-15
> **Author Response (1/3)**
>
> Thank you for your careful analysis of our work. We hope the following response addresses your concerns.
>
> **Clarifying design motivation in Section 4.**
>
> > The organization of Section 4 requires further refinement for improved readability. It would be beneficial to briefly outline the design motivation before delving into specific details, particularly regarding the content of Fig. 3.
>
> We apologize for any lack of clarity in Section 4. To improve clarity, we will briefly outline the motivation for our tamper-resistance loss at the top of Section 4.2. Thank you for your suggestion.
>
>
> **We have added mathematical formulations for the tamper-resistance losses.**
>
> > the mathematical formulation of the loss function used is currently absent.
>
> We kindly point the reviewer to Appendix B.2, where we fully describe the tamper-resistance losses used during TAR. However, we agree that adding mathematical formulas in the main paper for the tamper-resistance losses would improve readability. We have added these to the updated paper. Thank you for your suggestion.
>
>
> **Improved caption for Figure 1.**
>
> > The caption of figure 1 needs to explain the difference between the two branches more clearly. For example, what's the difference between the first nodes of the two branches.
>
> Please see below for the improved Figure 1 caption. We agree that the original caption lacked clarity, and we think the new version will help readers understand the problem setting better. Thank you for your suggestion.
>
> New caption:
> “An illustration comparing two approaches to LLM safety when subjected to adversarial fine-tuning. The top branch shows conventional safeguards (like refusal training), which can be easily bypassed when adversaries fine-tune the model weights to remove safety constraints. The bottom branch demonstrates our proposed TAR (Tamper-Resistant) method, which maintains robustness even when adversaries attempt to fine-tune the model to reintroduce harmful capabilities. While conventional safeguards can be circumvented by adversaries with access to model weights, TAR provides significantly more protection against attempts to weaponize open-weight LLMs through malicious fine-tuning.”
>
> **Clarifying 5000 step claim and attack configurations.**
>
> > In Section 1, the authors assert that the proposed method can endure fine-tuning of up to 5000 steps … this claim does not intuitively convey the contribution of the paper.
>
> The 5000-step attacks correspond to Adv. 1 and Adv. 2 in Table 9 and Figure 4, which utilize a fully held-out biology dataset. We agree that claims based on hyperparameters such as step-count could be misleading, and considering the hyperparameters of the attack is crucial. We show in Table 9 that the 5000-step adversaries use similar optimization hyperparameters as all other attacks in our red-teaming suite, with the learning rate for these 2 adversaries varied between $2\times10^{-5} $ and $ 4\times10^{-5}$. We believe our robustness results on these adversaries support our claims, in that our method is capable of withstanding much stronger optimization pressure compared to prior work. We will clarify this in the updated paper.
>
> > The details surrounding fine-tuning, such as batch size and learning rate, are unclear; these parameters significantly influence the number of fine-tuning steps.
>
> We have already fully specified these hyperparameters in Tables 7, 8, 9, and 10 in our Appendix.
>
> > Secondly, a comparative analysis with the typical number of steps required to fine-tune models on downstream tasks is lacking.
>
> In Table 9, we list out all of our attacks, which are performed on standard fine-tuning hyperparameters such as varying the learning between between $2\times10^{-6}$, $2\times10^{-5}$, and $4\times10^{-5}$, optimization steps of 1000 or 5000, and batch sizes of 32 or 64. We also consider LoRA training alongside full-parameter training. These are all representative of common fine-tuning settings for Llama-3-8B models.

---

> > ### Author Response · Authors · 2024-11-15
> > **Author Response (2/3)**
> >
> > **Clarifying the threat model and metrics**
> >
> > > The threat model lacks clarity. The authors assume that the attacker is compute-bounded but do not provide a clear definition of what this entails.
> >
> > When defining our threat model, our aim was to capture a broad range of attacks to define the scope of tamper-resistance research, rather than varying specific hyperparameters. As you pointed out, quantifying attacks strictly based on the number of optimization steps or other hyperparameters could be misleading. However, we agree that limiting the adversary’s compute budget with a concrete metric, such as total FLOPs, could bring more clarity to our threat model. Thank you for pointing this out.
> >
> > > Furthermore, including concrete examples of the metrics for capabilities_metric and safety_metric would enhance understanding.
> >
> > We include concrete examples of both the capabilities_metric and safety_metric in Section 3.2, in the two paragraphs immediately following the initial definition of these metrics. Specifically, we mention the use of MMLU and WMDP as our main evaluations for knowledge restriction, as well as MT-Bench and HarmBench for harmful request refusal. We would be happy to suggest other benchmarks for evaluations in this section, such as GSM8K or HumanEval. Thank you for this suggestion.
> >
> >
> > **Clarifying differences between tamper-resistance training and standard meta-learning.**
> >
> > > In section 4.2, the authors emphasized that the proposed method is different from standard meta-learning. However, the differences highlighted seem minor and do not present significant technical challenges.
> >
> > We agree that our regime lies within the meta-learning framework. In fact, we clearly acknowledge this several times in our paper. However, there are numerous substantial differences between standard use cases of meta-learning and our own, with significant technical implications:
> > - In standard meta-learning, the inner and outer loop optimizers typically use the same loss and are not adversarial. By contrast, our method uses adversarial inner and outer loop optimizers that are actively optimizing against each other. This adversarial optimization significantly affects the training dynamics.
> > - In standard meta-learning, the goal is to minimize the loss for test-time tasks in *as few steps as possible*. By contrast, in our threat model, the goal is to maintain high adversary loss at test-time for *as many steps as possible*. While both settings use the machinery of meta-learning, this is a fundamental difference with significant implications, one of which being that tamper-resistance training seeks to generalize robustness to a larger number of steps than seen at training time, which has no analogy in standard meta-learning.
> >
> > Due to the fundamental differences described above, we had to develop efficiency tricks to make training in this adversarial meta-learning possible with reasonable runtime, like our inner-loop subsampling trick in Appendix B.3 and B.4.
> >
> > We realize these differences are a bit in the weeds for most readers, so we will add a section to the appendix clarifying this distinction. Thank you for your suggestion.
> >
> >
> >
> >
> > **Theoretical analyses may be premature, so we instead follow best practices in empirical robustness research.**
> >
> > > The term 'empirically' is employed multiple times in the methods section. While drawing conclusions from empirical observations is valuable for informing solution design, relying solely on empirical data, particularly in the selection of a loss function, may impose limitations on the solution's robustness. A theoretical analysis comparing the efficacy of the entropy loss function versus cross-entropy loss is necessary.
> >
> > We agree that a theoretical analysis could be valuable. However, obtaining theoretical guarantees for tamper-resistance is quite difficult at this stage, given that research on the problem has only just begun. This is common in adversarial robustness research, where certified robustness often takes several years to catch up to empirical robustness.
> >
> > To provide a better understanding of the robustness of our method, we follow standard best practices in empirical robustness research. Namely, we adversarially stress-test our defense against a diverse suite of strong adversaries, including attacks specifically developed to break our defense (e.g., attacks using different training sets or optimizers than what we trained against). This led us to make several discoveries for how to improve empirical robustness, including the selection of a loss function that worked better across a broad range of test-time attacks. There is still much room for improvement on tamper-resistance, potentially including in the loss function design. Thus, a theoretical analysis of our particular loss function would likely be premature, since the methods are liable to change in followup work.

---

> > > ### Author Response · Authors · 2024-11-15
> > > **Author Response (3/3)**
> > >
> > > **Explanation of metrics used.**
> > >
> > > > The performance metrics used in the experiments require clear explanation.
> > >
> > > To clarify, we do not create new metrics. Rather, we use the existing MMLU and WMDP benchmarks—along with standard metrics for those benchmarks—to evaluate benign performance and safety properties. For the harmful request refusal experiments, we also use MT-Bench as a benign performance metric, and HarmBench to measure prompting ASR. We list the benchmarks and metrics for evaluation in Table 1 in its caption as well as in Section 3.2. We are happy to re-state the use of these benchmarks in Section 5.1 to improve clarity.
> > >
> > > **Capabilities-robustness tradeoff.**
> > >
> > > > Based on the results presented, the proposed solution appears to sacrifice task performance in favor of enhanced robustness, suggesting potential flaws in the method.
> > >
> > > We do observe a trade-off between benign capabilities and robustness, similar to nearly all prior work on adversarial robustness. While we would prefer to not have this trade-off, please note that this is not a technical flaw in the method. Rather, it is a common property of many adversarial robustness defenses, including ours.
> > >
> > > Previously, it was not possible to obtain tamper-resistance at all, even at much lower levels of benign performance. Our results show that with a moderate reduction in benign performance, we can greatly improve the tamper-resistance of LLMs, demonstrating for the first time that this problem is tractable. To reiterate, trading off benign performance is not a technical flaw in the method. However, we believe that improving benign capabilities alongside robustness is an important area for future work.
> > >
> > > Thank you again for your detailed comments. If we have addressed the thrust of your concerns, we kindly ask that you consider raising your score.

---

> > > > ### Author Response · Authors · 2024-11-25
> > > >
> > > > We would like to gently remind the reviewer that the rebuttal discussion period is ending soon, and that we are happy to answer any remaining concerns.

---

> ### Comment · Reviewer_vbKt · 2024-11-25
>
> Thanks for the authors' response that has addressed partial my doubts. Based on this, I have decided to modify my rating to 5.

---

### Official Review · Reviewer_rLai · 2024-11-03

**Soundness:** 2
**Presentation:** 3
**Contribution:** 2
**Rating:** 5
**Confidence:** 4

**Summary:**

This paper focuses on the issue of the lack of robustness in open LLMs when facing model weight tampering attacks. The authors propose a method called TAR, designed to establish tamper-resistant protection mechanisms for LLMs, ensuring that attackers cannot compromise these protections after minimal optimization. Extensive experiments validate the performance of this approach, showing that it outperforms existing methods.

**Strengths:**

The security issues related to open LLMs are both important and intriguing; The authors present a series of solutions to address these security threats; and experiments validate the performance of the proposed mechanisms.

**Weaknesses:**

1. Insufficient sustainability. This paper proposes the integration of adversarial learning and meta-learning to enhance the effectiveness of defense mechanisms, making it difficult for attackers to compromise them in a short period. However, this effectiveness actually depends on the diversity of attack types included in the training data for optimizing eqn.1. In other words, the resilience of the proposed mechanism may be superficial and does not guarantee the security of open-weight LLMs. Furthermore, the authors do not provide corresponding theoretical analysis or proof.
2. Incremental technical contributions. Although the paper is expressed clearly, its innovation is not evident in terms of both technical aspects and application scenarios. Specifically, the proposed solutions are based on existing widely used methods, and the authors have not clearly articulated their unique contributions. Therefore, it is recommended that the authors provide further clarification on this matter.
3. The performance of the proposed mechanism is closely related to the adversarial training methods and data, which means its resilience remains a significant issue.
4. The presentation of the performance comparison between TAR and existing mechanisms in Figure 5 is unclear and potentially confusing. The authors should provide further analysis of this result, explaining why the performance of TAR shows a significant change as the step size increases.

**Questions:**

1. Insufficient sustainability. This paper proposes the integration of adversarial learning and meta-learning to enhance the effectiveness of defense mechanisms, making it difficult for attackers to compromise them in a short period. However, this effectiveness actually depends on the diversity of attack types included in the training data for optimizing eqn.1. In other words, the resilience of the proposed mechanism may be superficial and does not guarantee the security of open-weight LLMs. Furthermore, the authors do not provide corresponding theoretical analysis or proof.
2. Incremental technical contributions. Although the paper is expressed clearly, its innovation is not evident in terms of both technical aspects and application scenarios. Specifically, the proposed solutions are based on existing widely used methods, and the authors have not clearly articulated their unique contributions. Therefore, it is recommended that the authors provide further clarification on this matter.
3. The performance of the proposed mechanism is closely related to the adversarial training methods and data, which means its resilience remains a significant issue.
4. The presentation of the performance comparison between TAR and existing mechanisms in Figure 5 is unclear and potentially confusing. The authors should provide further analysis of this result, explaining why the performance of TAR shows a significant change as the step size increases.

**Details Of Ethics Concerns:**

N/A.

---

> ### Author Response · Authors · 2024-11-15
> **Author Response**
>
> Thank you for your careful analysis of our work. We are glad you found our experiments to support the performance of our method. We hope the following response addresses your concerns.
>
> **Our main contribution is to show that tamper-resistance is tractable.**
>
> > Insufficient sustainability … this effectiveness actually depends on the diversity of attack types included in the training data … the resilience of the proposed mechanism may be superficial and does not guarantee the security of open-weight LLMs. Furthermore, the authors do not provide corresponding theoretical analysis or proof.
>
> In the paper, we clearly state that our method does not guarantee robustness across arbitrary adversaries. Obtaining theoretical guarantees for tamper-resistance is quite difficult at this stage, given that research on the problem has only just begun. To address this, we emphasize in the paper that extensive red teaming of methods is critical to gain confidence in their empirical robustness.
>
> For this reason, we stress-test our method with 28 diverse attacks, finding that some do succeed. Our main contribution is to show that tamper-resistance is a tractable problem in the first place, which we demonstrate for the first time in LLMs.
>
> **Clarifying technical contributions.**
>
> > Incremental technical contributions. Although the paper is expressed clearly, its innovation is not evident in terms of both technical aspects and application scenarios. Specifically, the proposed solutions are based on existing widely used methods, and the authors have not clearly articulated their unique contributions.
>
> Please see our above description of how our main contribution is to show that tamper-resistance for LLMs is tractable in the first place. This itself is a significant technical contribution that we stress throughout the paper.
>
> Our technical methods indeed build on prior work. As we mention in the paper, our TAR method builds on MLAC and first-order MAML. However, we make numerous significant technical contributions on top of these works as follows:
>
> Other technical contributions include:
> - Developing a tamper-resistance evaluation pipeline for LLMs, including an extensive red teaming suite. This did not exist prior to our work.
> - Demonstrating that MLAC does not scale to LLMs
> - Developing a 2-stage method with an initial unlearning/alignment stage that doesn’t use tamper-resistance training, which greatly outperforms prior work
> - Demonstrating that input-based adversarial training (e.g., R2D2) does not generalize to provide robustness to weight tampering attacks. This had not previously been shown.
> - Demonstrating that concurrent work (e.g., RepNoise) lacks robustness against a broader set of attacks
> - Identifying a significantly improved outer loop loss for tamper-resistance training
> - Introduced methods for minimizing memory overhead during meta-learning, making our adversarial meta-learning procedure possible.
>
> Given these contributions that can be found in our paper, we kindly disagree that the technical and experimental contributions of the paper are incremental. However, we agree that the contributions could be made more clear. We will add an explicit contributions list to the updated paper. Thank you for your suggestion.
>
> **We significantly improve resilience compared to prior work.**
>
> > The performance of the proposed mechanism is closely related to the adversarial training ... which means its resilience remains a significant issue.
>
> We agree that resilience to attacks is an important issue. As previously mentioned, our goal is to show that progress is possible in contrast with prior work, since the best existing methods are not robust to fine-tuning attacks. Thus, while our method does not provide full resilience, it does improve resilience.
>
> **Clarifying Figure 5.**
>
> > The presentation of the performance comparison between TAR and existing mechanisms in Figure 5 is unclear and potentially confusing. The authors should provide further analysis of this result, explaining why the performance of TAR shows a significant change as the step size increases.
>
> Please note that the optimizer step-size does not increase in Figure 5. Rather, the x-axis shows the number of optimizer steps. Also note that TAR keeps the adversary’s loss high (above the recovery region) across all 1000 steps, indicating that TAR’s performance is strong compared to the baseline.
>
> You may be asking about significant changes in the adversary’s loss shown in Figure 5. We partly explain these changes in Section 5.3. However, we agree that further analysis could improve clarity for readers, which we will add to the updated paper. Namely, we will explain that the model has implemented the entropy maximization by sharply increasing the cross-entropy loss at the beginning of fine-tuning. Thank you for your suggestion.
>
> Thank you again for your comments. If we have addressed the thrust of your concerns, we kindly ask that you consider raising your score.

---

> > ### Author Response · Authors · 2024-11-25
> >
> > We would like to gently remind the reviewer that the rebuttal discussion period is ending soon, and that we are happy to answer any remaining concerns.

---

> > ### Comment · Reviewer_rLai · 2024-11-26
> >
> > Thank you for your response. Some of my concerns have been resolved, and I will adjust my score accordingly.

---

### Official Review · Reviewer_Rc9Z · 2024-11-04

**Soundness:** 3
**Presentation:** 3
**Contribution:** 3
**Rating:** 8
**Confidence:** 3

**Summary:**

This paper focuses on the robustness of open-weight LLMs and proposes a novel defense method called TAR.

**Strengths:**

**About contribution**

+ The experimental results shown in Table 1 are significant enough to validate the main claims of this paper.
+ The proposed method is intuitive.  By providing detailed discussions of the related works, it is not hard to understand why the authors designed the algorithms as presented, even for readers not familiar with the defense of LLMs.

**About novelty**

According to Section 2, this paper proposes the first defense method for autoregressive LLMs against tampering attacks. To the best of my knowledge, concurrent jailbreaking attacks are mostly input-based. However, as claimed in Section 1, the tampering attacks are also posing threats to LLMs. This paper will bring new insight into the research on the robustness of LLMs.

**About presentation**

+ The preliminary part (Section 3) is brief and clear, making the technical part of this paper easy to follow.

**Weaknesses:**

**About presentation**

+ The authors do not discuss the cost of the experiments, including time cost and GPU memory cost. Section B.4 mentioned that the experiments use 8 A100 with 80GB GPU memory. What is the minimum requirement for the experiments?
+ I suggest including a statement of contribution to make this paper easier to follow.

**Questions:**

+ In Figure 2, the difference between TAR and the baseline methods is significant. However, it seems that the capabilities of TAR are lower than the baseline methods. Is there a trade-off between capabilities and tamper resistance?

**Details Of Ethics Concerns:**

No ethics review is needed.

---

> ### Author Response · Authors · 2024-11-19
> **Author Response**
>
> Thank you for your careful analysis of our work and positive rating. We are glad you found our method intuitive and that our experimental results validate our claims. We hope the following response addresses your concerns:
>
> **Runtime and memory costs.**
>
> > The authors do not discuss the cost of the experiments, including time cost and GPU memory cost. Section B.4 mentioned that the experiments use 8 A100 with 80GB GPU memory. What is the minimum requirement for the experiments?
>
> Our method, TAR, involves an inner-loop/outer-loop training setup, for conducting first-order meta-learning as described in both Algorithm 1 and Equation 1. For outer loop steps $N$ and inner loop steps $K$, the runtime complexity of the method is $O(NK)$. Generally, $K<<N$, as we observe empirically in Figure 6 in Appendix C.2 that inner-loop lengths of $K=64$ with an outer loop length of $N=750$, can already preserve a high adversary loss much more than $K$ steps.
>
> **Regarding run-time overhead**, for each inner loop step, both an adversary gradient and meta-gradient is computed for the adversary’s loss and tamper-resistance loss, respectively. The outer loop also computes one additional gradient for the retain-set, which is added to the accumulated meta-gradient. However, via the subsampling trick discussed in Appendix B.3, we compute meta-gradients every 4 adversary steps, which we found to speed up training by a constant factor without sacrificing robustness. In total, there are (750 retain gradients + 750\*64 adversary gradients + 750\*64/4 meta-gradients) = 60,750 forward/backward passes computed during our method with a batch size of 64.
>
> This is multiple orders of magnitude fewer forward/backward passes than used in original model pre-training, which is reported to be 1,200,000 optimization steps with a batch size of 8,000,000 for Llama-3-8B [1].
>
> **On 8 80GB A100 GPUs, our method with N=750 and K=64 can be run in 18 hours.**
>
> **Regarding memory overhead**, one might consider memory to be the primary concern for running TAR on large models. However, since we perform first-order meta-learning, as we explain in Appendix B.4, meta-gradients can be accumulated in-place in a separate data structure. Even with common model sharding algorithms such as FSDP, this can be performed without additional distributed communication, requiring additional memory equivalent to **only one extra copy of model gradients per device**. Thus, our method is quite memory-efficient as it scales memory usage by only a constant factor with sharded model size, which means **TAR has the same memory complexity as standard training asymptotically**.
>
> Thank you for bringing up this concern, we will include more discussion of these costs and our mitigations in our revision.
>
> [1] Llama Team, AI @ Meta. The llama 3 herd of models, 2024.
>
> **Contributions.**
>
> > I suggest including a statement of contribution to make this paper easier to follow.
>
> Thank you for your suggestion. We will do so in our revision. Specifically, we will highlight the following 3 core contributions:
> (1) We develop a tamper-resistance evaluation pipeline for LLMs, including an extensive red-teaming suite.
> (2) We show that tamper-resistance is tractable for LLMs, and we introduce the first method to provide robustness for a wide range of adversaries.
> (3) We show that existing methods for adversarial training of LLMs, including concurrently released unlearning methods, are not robust.
>
> **Capabilities-robustness tradeoff.**
>
> > In Figure 2, the difference between TAR and the baseline methods is significant. However, it seems that the capabilities of TAR are lower than the baseline methods. Is there a trade-off between capabilities and tamper resistance?
>
> We do observe a trade-off between benign capabilities and tamper-resistance, which we discuss in Section 5 and Appendix C.4. This is a similarity between our setting and standard adversarial robustness, e.g., for image classification, where adversarial training defenses often induce a trade-off between robustness and benign accuracy. Improving this tradeoff to reduce the cost of tamper-resistance defenses would be an interesting direction for future work.

---

### Official Review · Reviewer_n8EQ · 2024-11-04

**Soundness:** 2
**Presentation:** 3
**Contribution:** 4
**Rating:** 6
**Confidence:** 3

**Summary:**

The paper proposes a method (TAR) for improving tamper-resistance, i.e. model-level defense against adversarial finetuning attacks, of open-weight large language models. The method consists of several components: (1) initial safeguarding via _random mapping_ of harmful representations, (2) outer loop minimizing tamper-resistance and retain losses, (3) inner loop for computing tamper-resistance loss, which applies multiple finetuning attacks. Different design choices for tamper-resistance loss and its empirical significance are discussed: for weaponization knowledge restriction setting a negative _entropy_ loss is proposed, and for harmful request refusal a direct preference optimization (DPO) loss is used. The retain loss consists of language modeling loss and $l_2$-norm loss for representations of optimized and a base model. The results suggest the proposed method effectively defends the model against the majority of considered finetuning attacks, maintaining low accuracies on harmful questions post-(finetuning)-attack, although at the considerable cost of drop in accuracy on benign questions (pre-attack). Additionally, the authors acknowledge that the set of finetuning attacks during tamper-resistance training directly impacts the tamper-resistance against test-time attacks (e.g. "Retain $\rightarrow$ Forget" attack breaks the defense if it is not included in the training phase), suggesting the defense might struggle with unseen attacks (e.g. PEFT-attacks could break the defense in many of the settings).

**Strengths:**

1. **Significance** of the problem. The paper addresses an important and challenging problem of defending open-weight large language models against finetuning attacks. In authors words: "_This problem has been considered very challenging and by some intractable, as no method has yet provided substantial robustness to these attacks. However, making progress on this problem would provide a valuable tool to regulators and model developers by ameliorating the dual-use dilemma of open-weight models_".

2. **Originality**. Although most of the components of the proposed method are inspired from previous works (e.g. adversarial training, representation engineering), the overall approach constitutes substantial novel contribution to the field, to the best of my knowledge. The results on tamper-resistance for post-attack harmful accuracies demonstrate substantial improvement over previous methods (for most of the considered attacks).

**Weaknesses:**

1. **Evaluation** against out-of-distribution attacks.
- My main concern is that the defense might be effective mostly against observed attacks, and it could break against other unseen attacks. For example, Table 4 in Appendix shows that "Retain $\rightarrow$ Forget" attack breaks the defense if it is not included in the training phase. Figure 4, and Figure 8 from Appendix show that PEFT attacks are more effective than Full Parameter attacks (in case of Biosecurity, PEFT attacks break the proposed defense method), given that the TAR used Full Parameter attacks during training.
- Therefore, more emphasis and red teaming effort should be put into unseen attacks during evaluation, e.g. the Post-Attack scores in Table 1 could be divided into "in-distribution" and "out-of-distribution" attacks, where the categorization of attacks should be agnostic to LR, LR scheduler, optimizer, number of steps, batch size. In other words, out-of-distribution attacks could be defined as those that use fundamentally different approaches or data sources than what was used during training, rather than just different hyperparameters. Testing against attacks that use different optimization algorithms, loss functions, or data distributions not seen during training could provide a more comprehensive assessment of the method's robustness.
- Since PEFT finetuning is more compute-efficient than Full Parameter tuning, the variants of PEFT attacks with more optimization steps should be considered under the compute-constrained attacker setup. PEFT attacks should also be considered for harmful request refusal.
- Other out-of-distribution attacks could also be proposed and evaluated, e.g. initializing an attack by perturbing the weights into a random direction before following gradients to avoid local potentially gradient obfuscated region; or running a ground-truth informed attack by following a fixed worst-case direction towards the weights of a harmful model to observe how far the optimization should run to get harmful results.
- Input-level red teaming approaches (e.g. from HarmBench benchmark) could also be evaluated as alternative attacks, which do not include gradients or weight perturbations.


2. More **detailed analysis** of the method is missing. The problem of **obfuscated gradients** should be addressed.
- The results for Chemical Security in Table 1 suggest that post-attack harmful accuracies for TAR are lower than pre-attack ones, which is unexpected and worrying. Could you provide a more detailed analysis of this phenomenon? Could you investigate whether this is due to a quirk in your evaluation setup, or if it reveals something fundamental about how your method works?
- Also the plots in Figure 6 in Appendix show that the loss values first start to increase under the gradient-based attacks, which is surprising. Could the loss decrease just by switching the gradient sign in the beginning of the optimization? This might point towards the problem of obfuscated gradients [a]. Other attacks, e.g. gradient-free ones, or the exploration of the loss landscape could provide a better understanding of the phenomenon.
- Section A in Appendix states that post-attack accuracy on benign questions for TAR method is low. This should be reflected in the main paper, and the reasons for this phenomenon could be studied and discussed. Section C1 of Appendix addresses the problem of benign finetuning, however it does not provide comparison with benign finetuning of the base model (or other non-TAR trained harmless models). What percentage of harmful questions could appear in finetuning to prevent the benign learning? Could benign prompts from Over-Refusal benchmark [b] cause the issues with benign finetuning?
- Over-Refusal benchmark [b] results should be included for the restricted models for full evaluation.

3. The **capability-robustness tradeoff** could be studied and discussed more in-detail, since this is the main limiting factor of applying TAR comparing to baseline methods.
- From Table 1, TAR is considerably *worse than baselines in terms of benign capabilities* in adjacent domain knowledge for about 10% in all domains. What about other capabilities such as reasoning, multi-lingual understanding, creative writing, coding etc?
- Could the whole capabilities-robustness tradeoff curve be explored and demonstrated for TAR and for baseline methods by varying hyperparameters (e.g. such as $\lambda_{TR}$)? Could a single metric for the method's tradeoff performance be proposed and compared between baselines, similar to Accuracy-Robustness Tradeoff Score (ART-score) in [c]?

4. **Clarity**.
- Many important parts of the paper are in Appendix, e.g. Random Mapping, attack categories, many crucial evaluations (see above). This makes the main paper less clear and prevents it from being self-sufficient.
- Was a single model trained to be tamper-resistant against all 3 weaponization domains, or were 3 different TAR models considered (e.g. in Table 1)? Was a single model trained for both weaponization restriction and harmful request refusal, or 2 different ones? Could a single model be defended against all considered harms? How would it affect benign capabilities? Is the approach the same for baseline methods? It was not clear from the text for me. These details could be included in the experimental setup section, and you could include a diagram or table summarizing the model configurations used for different experiments.
- What are the computational costs of TAR comparing to benign PEFT, Full Parameter finetuning and other baselines?
- Minor comment: could the scores in Table 1 be scaled such that random model gets 0, and perfect model get 100? It would help visualizing the effectiveness of TAR more clearly.

[a] Athalye, A., Carlini, N., & Wagner, D. (2018, July). Obfuscated gradients give a false sense of security: Circumventing defenses to adversarial examples. In International conference on machine learning (pp. 274-283). PMLR.

[b] Cui, J., Chiang, W. L., Stoica, I., & Hsieh, C. J. (2024). OR-Bench: An Over-Refusal Benchmark for Large Language Models. arXiv preprint arXiv:2405.20947.

[c] Nurlanov, Z., Schmidt, F.R., Bernard, F. (2024). Adaptive Certified Training: Towards Better Accuracy-Robustness Tradeoffs. In: Bifet, A., et al. Machine Learning and Knowledge Discovery in Databases. Research Track and Demo Track. ECML PKDD 2024. Lecture Notes in Computer Science(), vol 14948. Springer, Cham. https://doi.org/10.1007/978-3-031-70371-3_8

**Questions:**

See weaknesses section for questions and suggestions. I would be happy to change my opinion if my main concerns regarding fair evaluation of the defense method could be addressed.

---

> ### Author Response · Authors · 2024-11-25
> **Author Response (1/6)**
>
> Thank you for your thorough analysis of our work. We are glad you found our paper to constitute a substantial novel contribution. We hope the following response addresses your concerns.
>
> **Evaluation of attacks.**
>
> > My main concern is that the defense might be effective mostly against observed attacks, and it could break against other unseen attacks. For example, Table 4 in Appendix shows that "Retain -> Forget" attack breaks the defense if it is not included in the training phase. Figure 4, and Figure 8 from Appendix show that PEFT attacks are more effective than Full Parameter attacks (in case of Biosecurity, PEFT attacks break the proposed defense method), given that the TAR used Full Parameter attacks during training.
>
> This is an important point, which we highlight in the paper. We discuss targeted patching of vulnerabilities in detail in Appendix C.3, finding that one can increase robustness to the Retain -> Forget adversary by including it at train-time.
>
> We do find and acknowledge that the robustness of the final model obtained by our method changes with the level of distribution shift in test-time adversaries. We refer to both Table 7 and Table 9, which show the various adversaries during train-time and test-time. We believe that obtaining the robustness we find in Figure 4 to many adversaries at test-time (Table 9) not included during train-time (Table 7), is a noteworthy result, since it’s not obvious that meta-learning methods should generalize to many different optimization hyperparameters. Furthermore, obtaining robustness to solely the train-time attacks is already a significant improvement from prior work.
>
> While it is true that ideal tamper-resistance algorithms should not require exhaustive sampling of all test-time attacks, we emphasize that we significantly improve robustness to weight-tampering attacks compared to prior work.
>
> **In-distribution vs out-of-distribution adversaries.**
>
> > Therefore, more emphasis and red teaming effort should be put into unseen attacks during evaluation, e.g. the Post-Attack scores in Table 1 could be divided into "in-distribution" and "out-of-distribution" attacks
>
> Thank you for this suggestion. We would like to point out that we use a much broader set of optimization attacks compared to concurrent work that we cite, with a much higher number of optimization steps. Our evaluation is by far the most rigorous conducted to date. We already include a wide variety of test-time optimization algorithms, including fundamentally different optimizers and different datasets, as listed in Table 9. We also vary standard optimization hyperparameters like learning rate and total step count to check robustness to these different distribution shifts.
>
> We are happy to separate our attack results into in-distribution and out-of-distribution attacks as you suggest, as a new table in the appendix in our revision. Regarding the distinction between these attacks, please see the next point.
>
> > … out-of-distribution attacks could be defined as those that use fundamentally different approaches or data sources than what was used during training, rather than just different hyperparameters. the categorization of attacks should be agnostic to LR, LR scheduler, optimizer, number of steps, batch size.
>
> Thank you for this interesting suggestion. We believe such categorization requires careful handling, since the definition of distribution shift in weight optimization is nuanced. Specifically, varying these hyperparameters (LR, scheduler, optimizer algorithm, # of steps, batch size), already constitutes distribution shifts that could result in significantly varying final adversary losses.
>
> We do consider many adversaries with varied optimizers to partially measure this, in Table 9.
>
> > Testing against attacks that use different optimization algorithms, loss functions, or data distributions not seen during training could provide a more comprehensive assessment of the method's robustness
>
> This is a great suggestion; we agree that including many more attacks is important for future red-teaming efforts. We primarily included attacks following the standard fine-tuning methodology to gauge whether obtaining robustness was at all possible. However, fundamentally different attack algorithms are important to evaluate. We discuss your follow-up suggestions and include results for them in the next points.

---

> > ### Author Response · Authors · 2024-11-25
> > **Author Response (2/6)**
> >
> > **Additional OOD attack results.**
> >
> > > Other out-of-distribution attacks could also be proposed and evaluated, e.g. initializing an attack by perturbing the weights into a random direction before following gradients to avoid local potentially gradient obfuscated region; or running a ground-truth informed attack by following a fixed worst-case direction towards the weights of a harmful model to observe how far the optimization should run to get harmful results.
> >
> > We strongly agree that future work should consider diverse adversaries. We do already consider adversaries with significant distribution shifts beyond just learning rate and number of optimization steps in Table 9 (e.g., the OOD dataset adversaries in Table 9, R->F adversaries, and many unseen optimizer algorithms).
> >
> > We’ve added the weight perturbation attack you proposed with the following formulation: for each parameter, we add the result of a normalized random direction multiplied by the corresponding layer’s weight norm multiplied by a perturbation scaling factor. We consider the layer’s original weight norm to ensure that perturbations are proportional to each layer. We consider the following perturbation scaling factors: [1e-5, 1e-4]. We then subject this model with perturbed weights to an SFT attack using Adv. 3’s hyperparameters. We observe the following post-attack accuracies for the Biosecurity domain:
> >
> > | Perturbation Scaling Factor | Post-Attack Accuracy |
> > |---------------------------|--------|
> > | 1e-5                 	| 24.7 	|
> > | 1e-4                 	| 24.6 	|
> >
> > The robustness of TAR remains stable even when we introduce perturbations to the model parameters before SFT. These results suggest that TAR's effectiveness doesn't depend on narrowly-defined gradient directions.
> >
> > Moreover, we propose an additional adversary that minimizes the summed loss on a batch of Retain-set examples along with a batch of Forget-set examples. Our goal for this attack is to evaluate whether TAR can generalize its defense to a mixture of data distributions, whose combination was unseen during training. Below, we present the post-attack accuracies for this adversary in the Biosecurity domain:
> >
> > | Learning Rate | Post-Attack Accuracy |
> > |--------------|------------------------------|
> > | 2e-5     	| 23.9                 	|
> > | 4e-5     	| 24.6                 	|
> >
> > The post-attack accuracies for this adversary contribute further evidence that TAR can generalize to test-time attacks beyond hyperparameter variations.
> >
> >
> > **Additional PEFT attacks.**
> >
> > > Since PEFT finetuning is more compute-efficient than Full Parameter tuning, the variants of PEFT attacks with more optimization steps should be considered under the compute-constrained attacker setup.
> >
> > We agree that further variations of PEFT attacks would be beneficial for further evaluation. Because our selected PEFT attacks already reach the ceiling of recoverable accuracy on weaponization knowledge at 1000 optimization steps, we hypothesized that running PEFT attacks for a longer duration would not reveal anything substantial about the failure modes of our method. However, we recognize the importance of representing the most potent attacks and so we have run the same PEFT attacks from Table 9 in our Appendix, for double the number of optimization steps (2000) for the Biosecurity domain. We can compare these values with the post-attack accuracies reported in the paper, comparing the optimization steps and WMDP-Bio post-attack scores in the following table:
> >
> > | LR | 1000 Steps Acc. | 2000 Steps Acc. |
> > |--------------|------------|------------|
> > | 2e-5 (Adv. 27)    	| 65.2    	| 68.0    	|
> > | 4e-5 (Adv. 28)    	| 64.7    	| 66.2    	|
> >
> > The minor increase in the post-attack increase for Adversary 27 at 2000 steps suggests that at 1000 steps, the PEFT adversary is already capable of recovering weaponization knowledge. Thus, we posit that PEFT adversaries do not benefit from longer optimization durations, even though they constitute a more compute-efficient SFT attack.
> >
> > >  PEFT attacks should also be considered for harmful request refusal.
> >
> > We have run a PEFT attack per your suggestion, using similar hyperparameters on the ToxicDPOv0.2 dataset as the test-time adversaries listed in Table 10 (batch size=32, 10 epochs, LR = $1\times10^{-5}$, and AdamW). The reported ASR is the HarmBench ASR, computed on the TAR-Refusal attacked via PEFT. We compare our model to the strongest baseline, Refusal-Trained Llama-3-8B.
> >
> > | Model    | PEFT Attack ASR |
> > | -------- | ------- |
> > | Refusal Trained  | 62.5    |
> > | TAR (Ours) | 62.9     |
> >
> > We find that PEFT constitutes a weaker attack for both methods in the harmful request refusal setting, as they do not achieve as high an ASR as full-param training on the baselines. Nonetheless, both TAR and Refusal-Trained Llama-3-8B recover to approximately the same ASR. As in the weaponization knowledge restriction results, improving robustness to more attacks, such as PEFT, is an important area for future work.

---

> ### Author Response · Authors · 2024-11-25
> **Author Response (3/6)**
>
> **Further explanation of our method.**
>
> > Also the plots in Figure 6 in Appendix show that the loss values first start to increase under the gradient-based attacks, which is surprising.
>
> We will be sure to improve our description of the loss dynamics in our method, to better motivate Figure 5 and 6. Specifically, the sharp increase in cross-entropy loss is not a symptom of numerical instability or other strictly unintended behavior, but rather a byproduct of how the tamper-resistance optimizer implements maximization of entropy during the adversary’s fine-tuning trajectory. We indeed observe that the tamper-resistance meta-learning procedure is working as intended via the red inner-loop curves in Figure 3 (the red curves are not smoothed), where the entropy of the posteriors of training adversaries increases smoothly throughout training.
>
> To reiterate the tamper-resistance objective, we state in Appendix B.2 that the tamper-resistance formulation we implement searches for model parameters $\theta$, such that after $K$ steps of fine-tuning $\theta$ on the cross-entropy loss on the forget set, *entropy* (the tamper-resistance loss) is still high. This is exactly what we observe in Figure 3; specifically, the entropy during each inner loop of TAR *increases* toward the maximum entropy value, while the adversary fine-tunes a cross-entropy loss in each inner loop. Because entropy and cross-entropy are related, a high entropy distribution will still result in a high cross-entropy loss with respect to the target token labels.
>
> To reduce confusion caused by the cross-entropy loss curve in Figure 5 and 6, we will update our revision to include a curve of the entropy of forget set posteriors during the same adversary cross-entropy loss curve, which shows the entropy being maximized during the early steps of the cross-entropy loss, thereby clearly illustrating that the initial increase in cross-entropy loss is a byproduct of the entropy maximization. Thank you for pointing this out.
>
> **Clarifying differences between our setting and traditional adversarial robustness.**
>
> > … This might point towards the problem of obfuscated gradients [a]. Other attacks, e.g. gradient-free ones, or the exploration of the loss landscape could provide a better understanding of the phenomenon.
>
> We apologize for any lack of clarity in our formulation. Please note that in Eq. 1, the `attack` applied to the model parameters $\theta$, *modifies the model parameters*. In traditional adversarial training, attacks are conducted on *model inputs*. Thus, our objective, while visually similar, is distinctly a meta-learning objective, since computing a gradient on our objective directly involves back-propagation through an inner-loop of fine-tuning.
>
> We wanted to draw a connection between our objective and adversarial training to bridge the gap for readers, since we do consider our method to be a *form* of adversarial training. However, we agree that this may introduce some confusion to the readers, as one might mistakenly assume that existing results in adversarial robustness for vision model inputs (e.g., obfuscated gradients) are directly applicable to our setting. We will improve our portrayal of Eq. 1 in the revision.
>
> Specifically, our method does not rely on numerical instability or other artifacts of obfuscated or shattered gradients from traditional adversarial robustness for vision to raise the adversary’s cross-entropy loss. We observe in red plots of Figure 3 (the red curves are not smoothed) that the entropy of the posteriors of training adversaries increases smoothly throughout training, which is clear evidence of the convergence of our first-order meta-learning objective as explained in the point above.
>
> **Negative gradient attack.**
>
> > Could the loss decrease just by switching the gradient sign in the beginning of the optimization?
>
> Thank you for suggesting this novel attack, which we implement as follows. We subjected TAR to this attack using the Adv. 3 (from Table 9) hyperparameter settings, while negating the gradient directions for the first 10, 50, and 100 optimization steps respectively. We summarize the results for the Biosecurity domain below:
>
> | # Initial Neg. Gradients | Post-Attack Acc. |
> |-----------------------------------|--------|
> | N = 10                        	| 24.0   |
> | N = 50                        	| 24.0   |
> | N = 100                       	| 24.7   |
>
> We find that the negative cross-entropy loss steadily decreases during the initial phase of gradient negation resulting in a large positive cross-entropy loss. Once the period of gradient negation ends, the loss decreases from its large, positive initial value, arriving at the same plateau shown for the blue “TAR Safeguard” curve in Figure 5. These results suggest to us that the negation of gradient directions which impede convergence to the pre-TAR, harmful basin, does not result in gradient directions which break the defense.

---

> ### Author Response · Authors · 2024-11-25
> **Author Response (4/6)**
>
> **Input-level red teaming.**
>
> > ​​Input-level red teaming approaches (e.g. from HarmBench benchmark) could also be evaluated as alternative attacks, which do not include gradients or weight perturbations.
>
> Thank you for this thoughtful suggestion. Combining input-level and parameter-level attacks is beyond the scope of our work but would be an interesting direction for future work. We will add a discussion of this to the updated paper. We do observe low pre-attack scores to weaponization knowledge questions in Table 1, in contrast with the high weaponization scores in the “No Defense” (Llama-3-8B-Instruct), suggesting a baseline level of input-space robustness.
>
>
> **Analysis of WMDP scores.**
>
> > The results for Chemical Security in Table 1 suggest that post-attack harmful accuracies for TAR are lower than pre-attack ones, which is unexpected and worrying. Could you provide a more detailed analysis of this phenomenon? Could you investigate whether this is due to a quirk in your evaluation setup, or if it reveals something fundamental about how your method works?
>
> Yes, we have observed, in some cases, that the attacker’s performance decreases post-attack. However, this is not a quirk in the evaluation setup. Rather, it reflects the adversarial training nature of the defense, where the defender is actively trying to reduce the attacker’s performance.
>
> > Could benign prompts from Over-Refusal benchmark [b] cause the issues with benign finetuning? Over-Refusal benchmark [b] results should be included for the restricted models for full evaluation.
>
> Thank you for this suggestion. While we were not able to collect the Over-Refusal benchmark results during the limited rebuttal window, we refer to our MT-Bench results presented in the next section as a proxy for over-refusal. These results illustrate that the model is still responsive to benign prompts.
>
> **Capabilities-robustness tradeoff.**
>
> > From Table 1, TAR is considerably worse than baselines in terms of benign capabilities in adjacent domain knowledge for about 10% in all domains.
>
> We do observe a trade-off between benign capabilities and tamper-resistance, which we discuss in Section 5 and Appendix C.4. This is a similarity between our setting and standard adversarial robustness, e.g., for image classification, where adversarial training defenses often induce a trade-off between robustness and benign accuracy. Improving this tradeoff to reduce the cost of tamper-resistance defenses would be an interesting direction for future work.
>
> > … What about other capabilities such as reasoning, multi-lingual understanding, creative writing, coding etc?
>
> Thank you for this suggestion. We’ve added to our evaluation of the weaponization restriction TAR model's capabilities by running the MT-Bench benchmark, which measures capabilities across a variety of benign task domains. Here are the results, comparing TAR in the Biosecurity domain with Llama 3 8B Instruct broken down by category:
>
>
> | Model            	| Turn	| Overall | Extraction | Math  | Humanities | Writing | Reasoning | STEM  | Roleplay | Coding |
> |---------------------|---------|---------|------------|-------|------------|---------|-----------|-------|----------|---------|
> | TAR (ours)      	| Turn 1  | 7.27	| 7.90   	| 5.80  | 7.90   	| 9.80	| 6.60  	| 7.70  | 8.40 	| 3.67	|
> | TAR (ours)      	| Turn 2  | 6.64	| 8.44   	| 4.00  | 9.43   	| 7.25	| 4.00  	| 8.67  | 8.00 	| 2.00	|
> | Llama 3 8B Instruct | Turn 1  | 8.34	| 9.10   	| 6.00  | 10.00  	| 10.00   | 7.20  	| 9.60  | 9.30 	| 5.50	|
> | Llama 3 8B Instruct | Turn 2  | 7.62	| 8.80   	| 4.40  | 9.90   	| 9.10	| 5.00  	| 9.50  | 8.90 	| 5.40	|
>
> In line with the observed capabilities-robustness tradeoff that we observe with MMLU, there is a tradeoff in scores for the capabilities you suggest (e.g., reasoning, writing, coding). Mitigating changes in capabilities while increasing robustness presents a meaningful direction for future work.
>
> > Could the whole capabilities-robustness tradeoff curve be explored and demonstrated for TAR and for baseline methods by varying hyperparameters (e.g. such as
> $\lambda_\text{TR}$)?
>
> Thank you for bringing this up. We agree that understanding the capabilities-robustness tradeoff in our method is important, as well as determining whether we can attribute the tradeoff to any particular hyperparameters.
>
> In Table 6 of Appendix C.4, we actually already discuss the effect of varying $\lambda_{TR}$ as you suggest. We found that increasing $\lambda_{TR}$ from $1.0$ to $4.0$, while holding $\lambda_{\text{retain}}$ at $1.0$, greatly improved overall tamper-resistance, in line with our expectations.

---

> > ### Author Response · Authors · 2024-11-25
> > **Author Response (5/6)**
> >
> > **Evaluating post-attack metrics.**
> >
> > > Section A in Appendix states that post-attack accuracy on benign questions for TAR method is low. This should be reflected in the main paper, and the reasons for this phenomenon could be studied and discussed. Section C1 of Appendix addresses the problem of benign finetuning, however it does not provide comparison with benign finetuning of the base model (or other non-TAR trained harmless models). What percentage of harmful questions could appear in finetuning to prevent the benign learning?
> >
> > Thank you for suggesting this highly relevant direction for analysis of TAR. In response to your request to diversify the types of attacks we subject our defense to, we present preliminary results for an adversary which minimizes the summed loss on a batch of Retain-set data along with a batch of Forget-set data, shown again below:
> >
> > | Learning Rate | Post-Attack Accuracy |
> > |--------------|------------------------------|
> > | 2e-5     	| 23.9                 	|
> > | 4e-5     	| 24.6                 	|
> >
> > These results suggest that, when presented in a consistent ratio to the defender, a one-to-one ratio of benign and harmful examples in a dataset prevents benign or Retain-set fine-tuning. Due to the computational cost of running attacks, we have not yet determined the exact ratio of harmful to benign examples that prevents benign fine-tuning. We defer this analysis to future work.
> >
> > **Tamper-resistance metric definition.**
> >
> > > Could a single metric for the method's tradeoff performance be proposed and compared between baselines, similar to Accuracy-Robustness Tradeoff Score (ART-score) in [c]?
> >
> > Thank you for this suggestion. Upon inspecting the ART-score discussed in [c], we believe that explicitly listed values for Retain and Forget accuracies are more interpretable for readers, since multiple-choice benchmark accuracies are easily understood. While combining them into a single metric could provide a single number for hill-climbing, this may introduce more indirection for readers, which we would prefer to minimize.
> >
> > **Computational cost of TAR.**
> >
> > > What are the computational costs of TAR comparing to benign PEFT, Full Parameter finetuning and other baselines?
> >
> > Each TAR fine-tuning run required 18 hours on 8 NVIDIA A100 80GB GPUs. This is approximately 8 orders of magnitude less compute than original pre-training for the Llama-3-8B model. For details, please see below.
> >
> > **Regarding run-time overhead**, for each inner loop step, both an adversary gradient and meta-gradient is computed for the adversary’s loss and tamper-resistance loss, respectively. The outer loop also computes one additional gradient for the retain-set, which is added to the accumulated meta-gradient. However, via the subsampling trick discussed in Appendix B.3, we compute meta-gradients every 4 adversary steps, which we found to speed up training by a constant factor without sacrificing robustness. In total, there are (750 retain gradients + 750*64 adversary gradients + 750*64/4 meta-gradients) = 60,750 forward/backward passes computed during our method with a batch size of 64.
> >
> > This is multiple orders of magnitude fewer forward/backward passes than used in original model pre-training, which is reported to be 1,200,000 optimization steps with a batch size of 8,000,000 for Llama-3-8B [1]. This comes out to a factor of around 10^8 fewer forward/backward passes for our TAR fine-tuning method compared to Llama-3-8B pre-training.
> >
> > **Regarding memory overhead**, one might consider memory to be the primary concern for running TAR on large models. However, since we perform first-order meta-learning, as we explain in Appendix B.4, meta-gradients can be accumulated in-place in a separate data structure. Even with common model sharding algorithms such as FSDP, this can be performed without additional distributed communication, requiring additional memory equivalent to **only one extra copy of model gradients per device.** Thus, our method is quite memory-efficient as it scales memory usage by only a constant factor with sharded model size, which means **TAR has the same memory complexity as standard training asymptotically.**
> >
> > All other baselines perform standard full-parameter training, only with modified loss functions. PEFT (e.g., LoRA) is more efficient than full-parameter training, since nearly all parameters are frozen, and low-rank adapters require considerably less memory to optimize.
> >
> > Thank you for bringing this up, we will include more discussion of these costs and our mitigations in our revision.
> >
> > [1] Llama Team, AI @ Meta. The llama 3 herd of models, 2024.

---

> ### Author Response · Authors · 2024-11-25
> **Author Response (6/6)**
>
> **Improving clarity.**
>
> We apologize for any confusion in our explanation of experiments. We clarify each of your questions below, and will update the paper in the revision to improve clarity.
>
> > Many important parts of the paper are in Appendix, e.g. Random Mapping, attack categories, many crucial evaluations (see above). This makes the main paper less clear and prevents it from being self-sufficient.
>
> We apologize for introducing any confusion. We wanted to prioritize discussion of the method and experimental results in the main text, which is limited to 10 pages. We will update our revision to clarify exactly where to look for further content in the Appendix, to improve the experience of the reader.
>
> > Was a single model trained to be tamper-resistant against all 3 weaponization domains, or were 3 different TAR models considered (e.g. in Table 1)? Was a single model trained for both weaponization restriction and harmful request refusal, or 2 different ones?
>
> The weaponization restriction and harmful request refusal are 2 separate settings; 3 separate models were trained for weaponization restriction (one model for each weaponization domain) and 1 model for harmful request refusal. We include the full details for each training setting in Appendices D.1 and D.2.
>
> > Could a single model be defended against all considered harms? How would it affect benign capabilities?
>
> This is a great question. For ease of evaluation and minimizing confounding variables, we opted to train different models for each setting. We were initially interested in seeing if achieving specificity of tamper-resistance to specific knowledge domains was possible, which we do observe in Table 1. It’s difficult to predict how benign capabilities would change as a result of defending against all harms in a single model. This would be a good direction to explore for future work.
>
> > Is the approach the same for baseline methods? It was not clear from the text for me. These details could be included in the experimental setup section, and you could include a diagram or table summarizing the model configurations used for different experiments.
>
> Yes; we implemented/reproduced the baseline methods to also individually restrict harms for each of the considered knowledge domains. We will update our revision to include a section that explicitly lists these configurations as you suggest. Thank you for pointing this out.
>
> > Minor comment: could the scores in Table 1 be scaled such that random model gets 0, and perfect model get 100? It would help visualizing the effectiveness of TAR more clearly.
>
> This is a great suggestion, since it could help readers understand performance on a normalized scale. Indeed, we do use this normalized tamper-resistance scaling for Figure 2, to show the relative tamper-resistance achieved for each domain. This is because the tamper-resistance achieved for each weaponization knowledge domain should be considered relative to the initial instruction-tuned/pretrain model’s baseline weaponization knowledge.
>
> For individual tables, such as Table 1, we believe multiple-choice QA accuracy is more legible for readers, since the performance relative to random chance (e.g., 25% for 4 possible choices) is commonly understood.
>
> Thank you again for your extensive analysis of our work. If we have addressed the thrust of your concerns, we kindly ask that you consider raising your score.

---

> > ### Comment · Reviewer_n8EQ · 2024-11-25
> > **Response to Authors**
> >
> > I appreciate the detailed author response.
> > 1. Although I still acknowledge the originality of the addressed problem, I am not convinced on applicability of the proposed method. The authors' response made it clear that TAR is computationally expensive (compared to finetuning), leads to substantially limited benign capabilities, and requires training separate TAR-models to defend against different harms.
> > 2. Second, I still have concerns about the evaluation.
> > - PEFT demonstrates strong attack performance, making TAR ineffective. Since PEFT hyperparameters can vary drastically, it could be challenging to include all PEFT-attacks into TAR-training.
> > - The loss function under gradient descent should decrease with a sufficiently small step size, i.e. for
> > $ \theta_{n+1}:= \theta_n - \gamma \nabla F(\theta_n) $, it is guaranteed that $F(\theta_{n+1}) \leq F(\theta_n)$ with a proper choice of $\gamma$. Therefore, I do not accept the authors' response that the increase in loss function (and decreased harmful performance for Chemical Security in Table 1) under the attack is an expected behavior for TAR optimized model. The reasons could be that the computed gradient is not correct, or the attack parameters are not properly tuned, or the method found a unique local optimum in high-dimensional space (although then the gradient should be 0). The proper analysis of the loss landscape post TAR-training was not performed, as requested.
> >
> > Therefore, I will maintain my score at 6, mainly due to originality of the problem, and I would recommend authors to remain cautious about the robustness claims.

---

> > > ### Author Response · Authors · 2024-12-03
> > >
> > > **Regarding the long-term viability of TAR given the strength of PEFT attacks**
> > >
> > > > PEFT demonstrates strong attack performance, making TAR ineffective. Since PEFT hyperparameters can vary drastically, it could be challenging to include all PEFT-attacks into TAR-training.
> > >
> > > While PEFT attacks currently break the TAR defense, we do not think that defending against PEFT attacks will present an insurmountable challenge to future work. Our experiments suggest that while PEFT hyperparameters can vary, the attacks still adhere to predictable optimization trajectories, very similar to those we observe under full parameter fine-tuning. We note that full parameter fine-tuning hyperparameters can also vary greatly, yet we were able to achieve significantly improved robustness against a range of these attacks. Most importantly, we included PEFT attacks as examples of OOD attacks **that we did not train against**. The current vulnerability to PEFT attacks is important, but it reflects the constraints of the current implementation, not a fundamental limitation in the approach. Although it is not yet a production-ready defense, TAR represents a meaningful step forward on this problem and warrants further research.
> > >
> > > **Response to expected behavior of a TAR-optimized model during harmful fine-tuning**
> > >
> > > > The loss function under gradient descent should decrease with a sufficiently small step size [...]
> > >
> > > To clarify, we find that the cross-entropy loss does not always increase over the first few steps of harmful fine-tuning. Notably, for learning rates 2e-06 through 4e-05 during Weaponization Chemical Security fine-tuning, the loss starts decreasing immediately and maintains a non-increasing trajectory. This optimization behavior substantiates that TAR does not depend on initial spikes in the cross-entropy loss to succeed. We apologize for any lack of clarity in our initial response and appreciate the reviewer's attention to this detail.
> > >
> > > The reviewer suggests that uncalibrated attack parameters may explain initial increases in the loss during harmful fine-tuning. However, we note that these same attack configurations successfully recover performance against baseline defenses. The fact that TAR resists attacks that succeed on other defenses is itself a demonstration of robustness.
> > >
> > > We would like to thank you again for your engagement and constructive feedback. If our response has addressed your main concerns, we kindly request that you consider raising your score.

---

### Public Comment · ~Tiansheng_Huang1 · 2024-11-26
**My two cents**

Hi,

I am a researcher who is actively working on the harmful fine-tuning problem that this paper aims to solve. I carefully read the paper and the review, and from the review, I think the main concern lies in the technical contribution of the proposed solution. I would like to express my viewpoints I have on TAR's technical design.

* TAR main component heavily lies in the tamper-resistance loss term, which very much resembles the loss term which is extensively used in meta-learning, and this term is optimized also through a first-order approximation like Maml, causing concern of its novelty.

* With that said, I think the main contribution of this paper is not to propose completely new and fancy solutions to solve an existing problem, its main contribution is to **propose a usable and interpretable solution** to solve the newly arising problem for LLMs fine-tuning and deployment. The application scenario and the problem being considered are different from that of meta-learning. I also noticed that the authors do make a lot of effort in making further optimization of the algorithms to adapt to the new problem context.

I personally feel that this is one of the papers that I would like to visit at the conference, therefore I would like to express my appreciation here.

Thanks,

Tiansheng Huang

https://huangtiansheng.github.io/

---

### Meta-Review · Area_Chair_wzDF · 2024-12-09

**Metareview:**

The paper presents a model-level defense method against adversarial finetuning attacks, called TAR.  According to the reviews, reviewers primarily raised concerns about the lack of technical novelty. After considering the authors' response, I think these concerns should not hinder this paper's acceptance. The core contribution lies in providing a specific solution to improve the robustness of LLMs to parameter-based attacks. In light of the positive average rating, I recommend accepting the paper.

**Additional Comments On Reviewer Discussion:**

Reviewers raised several concerns about the technical contribution, performance against unseen attacks, and computational costs. The authors provided detailed explanations and analyses that addressed most of these issues. However, Reviewer i3oQ still maintains concerns about the core contribution.

---

### Decision · Program_Chairs · 2025-01-22

Accept (Poster)